# GREAT Score: Global Robustness Evaluation of Adversarial Perturbation using Generative Models

**ZAITANG LI**
*The Chinese University of Hong Kong*
*Sha Tin, Hong Kong*
ztli@cse.cuhk.edu.hk

Pin-Yu Chen
*IBM Research*
*New York, USA*
pin-yu.chen@ibm.com

Tsung-Yi Ho
*The Chinese University of Hong Kong*
*Sha Tin, Hong Kong*
tyho@cse.cuhk.edu.hk

## Abstract

Current studies on adversarial robustness mainly focus on aggregating *local* robustness results from a set of data samples to evaluate and rank different models. However, the local statistics may not well represent the true *global* robustness of the underlying unknown data distribution. To address this challenge, this paper makes the first attempt to present a new framework, called GREAT Score, for global robustness evaluation of adversarial perturbation using generative models. Formally, GREAT Score carries the physical meaning of a global statistic capturing a mean certified attack-proof perturbation level over all samples drawn from a generative model. For finite-sample evaluation, we also derive a probabilistic guarantee on the sample complexity and the difference between the sample mean and the true mean. GREAT Score has several advantages: (1) Robustness evaluations using GREAT Score are efficient and scalable to large models, by sparing the need of running adversarial attacks. In particular, we show high correlation and significantly reduced computation cost of GREAT Score when compared to the attack-based model ranking on RobustBench [12]. (2) The use of generative models facilitates the approximation of the unknown data distribution. In our ablation study with different generative adversarial networks (GANs), we observe consistency between global robustness evaluation and the quality of GANs. (3) GREAT Score can be used for remote auditing of privacy-sensitive black-box models, as demonstrated by our robustness evaluation on several online facial recognition services.

---

**Project Demo and Code Page:**

https://huggingface.co/spaces/TrustSafeAI/GREAT-Score
https://github.com/IBM/GREAT-Score

---

## 1 Introduction

Adversarial robustness is the study of model performance in the worst-case scenario, which is a key element in trustworthy machine learning. Adversarial robustness evaluation refers to the process of assessing a model's resilience against adversarial attacks, which are inputs intentionally designed to deceive the model. Without further remediation, state-of-the-art machine learning models, especially neural networks, are known to be overly sensitive to small human-imperceptible perturbations to data

38th Conference on Neural Information Processing Systems (NeurIPS 2024).

inputs [19]. Such a property of over-sensitivity could be exploited by bad actors to craft adversarial perturbations leading to prediction-evasive adversarial examples.

Given a threat model specifying the knowledge of the target machine learning model (e.g., white-box or black-box model access) and the setting of plausible adversarial interventions (e.g., norm-bounded input perturbations), the methodology for adversarial robustness evaluation can be divided into two categories: *attack-dependent* and *attack-independent*. Attack-dependent approaches aim to devise the strongest possible attack and use it for performance assessment. A typical example is Auto-Attack [10], a state-of-the-art attack based on an ensemble of advanced white-box and black-box adversarial perturbation methods. On the other hand, attack-independent approaches aim to develop a certified or estimated score for adversarial robustness, reflecting a quantifiable level of attack-proof certificate. Typical examples include neural network verification techniques [60, 66], certified defenses such as randomized smoothing [9], and local Lipschitz constant estimation [59].

Despite a plethora of adversarial robustness evaluation methods, current studies primarily focus on aggregating *local* robustness results from a set of data samples. However, the sampling process of these test samples could be biased and unrepresentative of the true *global* robustness of the underlying data distribution, resulting in the risk of incorrect or biased robustness benchmarks. For instance, we find that when assessing the ranking of Imagenet models through Robustbench [11], using AutoAttack [10] with 10,000 randomly selected samples (the default choice) with 100 independent trials results in an unstable ranking coefficient of $0.907\pm0.0256$ when compared to that of the entire 50,000 test samples. This outcome affirms that AutoAttack's model ranking has notable variations with an undersampled or underrepresented test dataset.

An ideal situation is when the data distribution is transparent and one can draw an unlimited number of samples from the true distribution for reliable robustness evaluation. But in reality, the data distribution is unknown and difficult to characterize. In addition to lacking rigorous global robustness evaluation, many attack-independent methods are limited to the white-box setting, requiring detailed knowledge about the target model (e.g., model parameters and architecture) such as input gradients and internal data representations for robustness evaluation. Moreover, state-of-the-art attack-dependent and attack-independent methods often face the issue of scalability to large models and data volumes due to excessive complexity, such as the computational costs in iterative gradient computation and layer-wise interval bound propagation and relaxation [20, 31].

To address the aforementioned challenges including (i) lack of proper global adversarial robustness evaluation, (ii) limitation to white-box settings, and (iii) computational inefficiency, in this paper we present a novel attack-independent evaluation framework called *GREAT Score*, which is short for global robustness evaluation of adversarial perturbation using generative models. We tackle challenge (i) by using a generative model such as a generative adversarial network (GAN) [17, 18] or a diffusion model [27] as a proxy of the true unknown data distribution. Formally, GREAT Score is defined as the mean of a certified lower bound on minimal adversarial perturbation over the data sampling distribution of a generative model, which represents the global distribution-wise adversarial robustness with respect to the generative model in use. It entails a global statistic capturing the mean certified attack-proof perturbation level over all samples from a generative model. For finite-sample evaluation, we also derive a probabilistic guarantee quantifying the sample complexity and the difference between the sample mean and true mean.

For challenge (ii), our derivation of GREAT Score leads to a neat closed-form solution that only requires data forward-passing and accessing the model outputs, which applies to any black-box classifiers giving class prediction confidence scores as model output. Moreover, as a byproduct of using generative models, our adversarial robustness evaluation procedure is executed with only synthetically generated data instead of real data, which is particularly appealing to privacy-aware robustness assessment schemes, e.g., remote robustness evaluation or auditing by a third party with restricted access to data and model. We will present how GREAT Score can be used to assess the robustness of online black-box facial recognition models. Finally, for challenge (iii), GREAT Score is applicable to any off-the-self generative models so that we do not take the training cost of generative models into consideration. Furthermore, the computation of GREAT Score is lightweight because it scales linearly with the number of data samples used for evaluation, and each data sample only requires one forward pass through the model to obtain the final predictions.

We highlight our main contributions as follows:

- We present GREAT Score as a novel framework for deriving a global statistic representative of the distribution-wise robustness to adversarial perturbation, based on an off-the-shelf generative model for approximating the data generation process.

- Theoretically, we show that GREAT Score corresponds to a mean certified attack-proof level of $\mathcal{L}_2$-norm bounded input perturbation over the sampling distribution of a generative model (Theorem 1). We further develop a formal probabilistic guarantee on the quality of using the sample mean as GREAT Score with a finite number of samples from generative models (Theorem 2).

- We evaluate the effectiveness of GREAT Score on all neural network models on RobustBench [11] (the largest adversarial robustness benchmark), with a total of 17 models on CIFAR-10 and 5 models on ImageNet. We show that the model ranking of GREAT Score is highly aligned with that of the original ranking on RobustBench using AutoAttack [10], while GREAT Score significantly reduces the computation time. Specifically, on CIFAR-10 the computation complexity can be reduced by up to 2,000 times. The results suggest that GREAT Score is a competitive and computationally-efficient approach complementary to attack-based robustness evaluations.

- As a demonstration of GREAT Score's capability for remote robustness evaluation of access-limited systems, we show how GREAT Score can audit several online black-box facial recognition APIs.

## 2 Background and Related Works

**Adversarial Attack and Defense.** Adversarial attacks aim to generate examples that can evade classifier predictions in classification tasks. In principle, adversarial examples can be crafted by small perturbations to a native data sample, where the level of perturbation is measured by different $\mathcal{L}_p$ norms [7, 8, 58]. The procedure of finding adversarial perturbation within a perturbation level is often formulated as a constrained optimization problem, which can be solved by algorithms such as projected gradient descent (PGD) [38]. The state-of-the-art adversarial attack is the Auto-Attack [10], which uses an ensemble of white-box and black-box attacks. There are many methods (defenses) to improve adversarial robustness. A popular approach is adversarial training [38], which generates adversarial perturbation during model training for improved robustness. One common evaluation metric for adversarial robustness is robust accuracy, which is defined as the accuracy of correct classification under adversarial attacks, evaluated on a set of data samples. RobustBench [10] is the largest-scale standardized benchmark that ranks the models using robust accuracy against Auto-Attack on test sets from image classification datasets such as CIFAR-10 . In addition to discussed works, several studies evaluate model robustness differently. [43] introduce adversarial sparsity, quantifying the difficulty of finding perturbations, providing insights beyond adversarial accuracy. [48] propose probabilistic robustness, balancing average and worst-case performance by enforcing robustness to most perturbations, better addressing trade-offs. [22] introduce the adversarial hypervolume metric, a comprehensive measure of robustness across varying perturbation intensities.

**Generative Models.** Statistically speaking, let $X$ denote the observable variable and let $Y$ denote the corresponding label, the learning objective for a generative model is to model the conditional probability distribution $P(X \mid Y)$. Among all the generative models, GANs have gained a lot of attention in recent years due to their capability to generate realistic high-quality images [18]. The principle of training GANs is based on the formulation of a two-player zero-sum min-max game to learn the high-dimension data distribution. Eventually, these two players reach the Nash equilibrium that $D$ is unable to further discriminate real data versus generated samples. This adversarial learning methodology aids in obtaining high-quality generative models. In practice, the generator $G(\cdot)$ takes a random vector $z$ (i.e., a latent code) as input, which is generated from a zero-mean isotropic Gaussian distribution denoted as $z \sim \mathcal{N}(0, I)$, where $I$ means an identity matrix. Conditional GANs refer to the conditional generator $G(\cdot|Y)$ given a class label $Y$. In addition to GAN, diffusion models (DMs) are also gaining popularity. DMs consist of two stages: the forward diffusion process and the reverse diffusion process. In the forward process, the input data is gradually perturbed by Gaussian Noises and becomes an isotropic Gaussian distribution eventually. In the reverse process, DMs reverse the forward process and implement a sampling process from Gaussian noises to reconstruct the true samples by solving a stochastic differential equation. In our proposed framework, we use off-the-shelf (conditional) GANs and DMs (e.g., DDPM [27]) that are publicly available as our generative models.

**Formal Local Robustness Guarantee and Estimation.** Given a data sample $x$, a formal local robustness guarantee refers to a certified range on its perturbation level such that within which the top-1 class prediction of a model will remain unchanged [26]. In $\mathcal{L}_p$-norm ($p \geq 1$) bounded perturbations

centered at $x$, such a guarantee is often called a certified radius $r$ such that any perturbation $\delta$ to $x$ within this radius (i.e., $\|\delta\|_p \leq r$) will have the same top-1 class prediction as $x$. Therefore, the model is said to be provably locally robust (i.e., attack-proof) to any perturbations within the certified radius $r$. By definition, the certified radius of $x$ is also a lower bound on the minimal perturbation required to flip the model prediction.

Among all the related works on attack-independent local robustness evaluations, the CLEVER framework proposed in [59] is the closest to our study. The authors in [59] derived a closed-form of certified local radius involving the maximum local Lipschitz constant of the model output with respect to the data input around a neighborhood of a data sample $x$. They then proposed to use extreme value theory to estimate such a constant and use it to obtain a local robustness score, which is not a certified local radius. Our proposed GREAT Score has major differences from [59] in that our focus is on global robustness evaluation, and our GREAT Score is the mean of a certified radius over the sampling distribution of a generative model. In addition, for every generated sample, our local estimate gives a certified radius.

**Notations.** All the main notations used in the paper are summarized in Appendix A.

## 3 GREAT Score: Methodology and Algorithms

### 3.1 True Global Robustness and Certified Estimate

Let $f = [f_1, \ldots, f_K] : \mathbb{R}^d \to \mathbb{R}^K$ denote a fixed $K$-way classifier with flattened data input of dimension $d$, $(x, y)$ denote a pair of data sample $x$ and its corresponding groundtruth label $y \in \{1, \ldots, K\}$, $P$ denote the true data distribution which in practice is unknown, and $\Delta_{\min}(x)$ denote the minimal perturbation of a sample-label pair $(x, y) \sim P$ causing the change of the top-1 class prediction such that $\arg\max_{k \in \{1,\ldots,K\}} f_k(x + \Delta_{\min}(x)) \neq \arg\max_{k \in \{1,\ldots,K\}} f_k(x)$. Note that if the model $f$ makes an incorrect prediction on $x$, i.e., $y \neq \arg\max_{k \in \{1,\ldots,K\}} f_k(x)$, then we define $\Delta_{\min}(x) = 0$. This means the model is originally subject to prediction evasion on $x$ even without any perturbation. A higher $\Delta_{\min}(x)$ means better local robustness of $f$ on $x$.

The following statement defines the true global robustness of a classifier $f$ based on the probability density function $p(\cdot)$ of the underlying data distribution $P$.

**Definition 1** (True global robustness **w.r.t.** $P$). The true global robustness of a classifier $f$ with respect to a data distribution $P$ is defined as:

$$\Omega(f) = \mathbb{E}_{x \sim P}[\Delta_{min}(x)] = \int_{x \sim P} \Delta_{\min}(x) p(x) dx \tag{1}$$

Unless the probability density function of $P$ and every local minimal perturbation are known, the exact value of the true global robustness cannot be computed. An alternative is to estimate such a quantity. Extending Definition 1, let $g(x)$ be a local robustness statistic. Then the corresponding global robustness estimate is defined as

$$\widehat{\Omega}(f) = \mathbb{E}_{x \sim P}[g(x)] = \int_{x \sim P} g(x) p(x) dx \tag{2}$$

Furthermore, if one can prove that $g(x)$ is a valid lower bound on $\Delta_{min}(x)$ such that $g(x) \leq \Delta_{min}(x), \forall x$, then the estimate $\widehat{\Omega}(f)$ is said to be a certified lower bound on the true global robustness with respect to $P$, and larger $\widehat{\Omega}(f)$ will imply better true global robustness. In what follows, we will formally introduce our proposed GREAT Score and show that it is a certified estimate of the lower bound on the true robustness with respect to the data-generating distribution learned by a generative model.

### 3.2 Using GMs to Evaluate Global Robustness

Recall that a generative model (GM) takes a random vector $z \sim \mathcal{N}(0, I)$ sampled from a zero-mean isotropic Gaussian distribution as input to generate a data sample $G(z)$. In what follows, we present our first main theorem that establishes a certified lower bound $\widehat{\Omega}(f)$ on the true global robustness of a classifier $f$ measured by the data distribution given by $G(\cdot)$.

Without loss of generality, we assume that all data inputs are confined in the scaled data range $[0, 1]^d$, where $d$ is the size of any flattened data input. The $K$-way classifier $f : [0, 1]^d \mapsto \mathbb{R}^K$ takes a data sample $x$ as input and outputs a $K$-dimensional vector $f(x) = [f_1(x), \ldots, f_K(x)]$ indicating the likelihood of its prediction on $x$ over $K$ classes, where the top-1 class prediction is defined as $\hat{y} = \arg\max_{k=\{1,\ldots,K\}} f_k(x)$. We further denote $c$ as the groundtruth class of $x$. Therefore, if $\hat{y} \neq c$, then the classifier is said to make a wrong top-1 prediction. When considering the adversarial robustness on a wrongly classified sample $x$, we define the minimal perturbation for altering model prediction as $\Delta_{\min}(x) = 0$. The intuition is that an attacker does not need to take any action to make the sample $x$ evade the correct prediction by $f$, and therefore the required minimal adversarial perturbation level is 0 (i.e., zero robustness).

Given a generated data sample $G(z)$, we now formally define a local robustness score function as

$$g\left(G(z)\right) = \sqrt{\frac{\pi}{2}} \cdot \max\{f_c(G(z)) - \max_{k \in \{1,\ldots,K\}, k \neq c} f_k(G(z)), 0\} \tag{3}$$

The scalar $\sqrt{\pi/2}$ is a constant associated with the sampling Gaussian distribution of $G$, which will be apparent in later analysis. We further offer several insights into understanding the physical meaning of the considered local robustness score in (3): (i) The inner term $f_c(G(z)) - \max_{k \in \{1,\ldots,K\}, k \neq c} f_k(G(z))$ represents the gap in the likelihood of model prediction between the correct class $c$ and the most likely class other than $c$. A positive and larger value of this gap reflects higher confidence of the correct prediction and thus better robustness. (ii) Following (i), a negative gap means the model is making an incorrect prediction, and thus the outer term $\max\{\text{gap}, 0\} = 0$, which corresponds to zero robustness.

Next, we use the local robustness score $g$ defined in (3) to formally state our theorem on establishing a certified lower bound on the true global robustness and the proof sketch.

**Theorem 1** (certified global robustness estimate)**.** *Let $f : [0, 1]^d \mapsto [0, 1]^K$ be a $K$-way classifier and let $f_k(\cdot)$ be the predicted likelihood of class $k$, with $c$ denoting the groundtruth class. Given a generator $G$ such that it generates a sample $G(z)$ with $z \sim \mathcal{N}(0, I)$. Define $g\left(G(z)\right) = \sqrt{\frac{\pi}{2}} \cdot \max\{f_c(G(z)) - \max_{k \in \{1,\ldots,K\}, k \neq c} f_k(G(z)), 0\}$. Then the global robustness estimate of $f$ evaluated with $\mathcal{L}_2$-norm bounded perturbations, defined as $\widehat{\Omega}(f) = \mathbb{E}_{z \sim \mathcal{N}(0,I)}[g(G(z))]$, is a certified lower bound of the true global robustness $\Omega(f)$ with respect to $G$.*

The complete proof is given in Appendix C.

## 3.3 Probabilistic Guarantee on Sample Mean

As defined in Theorem 1, the global robustness estimate $\widehat{\Omega}(f) = \mathbb{E}_{z \sim \mathcal{N}(0,I)}[g(G(z))]$ is the mean of the local robustness score function introduced in (3) evaluated through a generator $G$ and its sampling distribution. In practice, one can use a finite number of samples $\{G(z_i|y_i)\}_{i=1}^n$ generated from a conditional generator $G(\cdot|y)$ to estimate $\widehat{\Omega}(f)$, where $y$ denotes a class label and it is also an input parameter to the conditional generator. The simplest estimator of $\widehat{\Omega}(f)$ is the sample mean, defined as

$$\widehat{\Omega}_S(f) = \frac{1}{n} \sum_{i=1}^n g(G(z_i|y_i)) \tag{4}$$

In what follows, we present our second main theorem to deliver a probabilistic guarantee on the sample complexity to achieve $\epsilon$ difference between the sample mean $\widehat{\Omega}_S(f)$ and the true mean $\widehat{\Omega}(f)$.

**Theorem 2** (probabilistic guarantee on sample mean)**.** *Let $f$ be a $K$-way classifier with its outputs bounded by $[0, 1]^K$ and let $e$ denote the natural base. For any $\epsilon, \delta > 0$, if the sample size $n \geq \frac{32e \cdot \log(2/\delta)}{\epsilon^2}$, then with probability at least $1 - \delta$, the sample mean $\widehat{\Omega}_S(f)$ is $\epsilon$-close to the true mean $\widehat{\Omega}(f)$. That is, $|\widehat{\Omega}_S(f) - \widehat{\Omega}(f)| \leq \epsilon$.*

The complete proof is given in Appendix D. The proof is built on a concentration inequality in [40]. It is worth noting that the bounded output assumption of the classifier $f$ in Theorem 2 can be easily satisfied by applying a normalization layer at the final model output, such as the softmax function or the element-wise sigmoid function.

## 3.4 Algorithm and Computational Complexity

Algorithm 1 summarizes the procedure of computing GREAT Score using the sample mean estimator. It can be seen that the computation complexity of GREAT Score is linear in the number of generated samples $N_S$, and for each sample, the computation of the statistic $g$ defined in (3) only requires drawing a sample from the generator $G$ and taking a forward pass to the classifier $f$ to obtain the model predictions on each class. As a byproduct, GREAT Score applies to the setting when the classifier $f$ is a black-box model, meaning only the model outputs are observable by an evaluator.

---

**Algorithm 1:** GREAT Score Computation

**Input:** $K$-way classifier $f(\cdot)$, conditional generator $G(\cdot)$, local score function $g(\cdot)$ defined in (3), number of generated samples $N_S$

**Output:** GREAT Score $\widehat{\Omega}_S(f)$

**for** $i \leftarrow 1$ to $N_S$ **do**

    Randomly select a class label $y \in \{1, 2, \dots, K\}$

    Sample $z \sim \mathcal{N}(0, I)$ from a Gaussian distribution and generate a sample $G(z|y)$ with class $y$

    Pass $G(z|y)$ into the model $f$ and get the prediction for each class $\{f_k(G(z|y))\}_{k=1}^K$

    Record the statistic

$$g^{(i)}(G(z|y)) = \sqrt{\frac{\pi}{2}} \cdot \max\{f_y(G(z|y)) - \max_{k \in \{1,\dots,K\},\, k \neq y} f_k(G(z|y)), 0\}$$

**end**

$\widehat{\Omega}_S(f) \leftarrow$ Compute the sample mean of $\{g^{(i)}\}_{i=1}^{N_S}$

---

## 3.5 Calibrated GREAT Score

In cases when one has additional knowledge of adversarial examples on a set of images from a generative model, e.g., successful adversarial perturbations (an upper bound on the minimal perturbation of each sample) returned by any norm-minimization adversarial attack method such as the CW attack [7], the CW attack employs two loss terms, classification loss and distance metric, to generate adversarial examples. See Appendix E for details. We can further "calibrate" the GREAT Score with respect to the available perturbations. Moreover, since Theorem 1 informs some design choices on the model output layer, as long as the model output is a non-negative $K$-dimensional vector $f \in [0,1]^K$ reflecting the prediction confidence over $K$ classes, we will incorporate such flexibility in the calibration process.

Specifically, we use calibration in the model ranking setup where there are $M$ models $\{f^{(j)}\}_{j=1}^M$ for evaluation, and each model (indexed by $j$) has a set of known perturbations $\{\delta_i^{(j)}\}_{i=1}^N$ on a common set of $N$ image-label pairs $\{x_i, y_i\}_{i=1}^N$ from the same generative model. We further consider four different model output layer designs (that are attached to the model logits): (i) sigmoid($\cdot|T_1$): sigmoid with temperature $T_1$, (ii) softmax($\cdot|T_2$): softmax with temperature $T_2$, (iii) sigmoid(softmax($\cdot|T_2 = 1)|T_1$): sigmoid with temperature after softmax, and (iv) softmax(sigmoid($\cdot|T_1 = 1)|T_2$): softmax with temperature after sigmoid. Finally, let $\{\widehat{\Omega}_S(f^{(j)})\}_{j=1}^M$ denote the GREAT Score computed based on $\{x_i, y_i\}_{i=1}^N$ for each model. We calibrate GREAT Score by optimizing some rank statistics (e.g., the Spearman's rank correlation coefficient) over the temperature parameter by comparing the ranking consistency between $\{\widehat{\Omega}_S(f^{(j)})\}_{j=1}^M$ and $\{\delta_i^{(j)}\}_{i=1}^N$. In our experiments, we find that setting (iv) gives the best result and use it as the default setup for calibration, as detailed in Appendix F.

# 4 Experimental Results

## 4.1 Experiment Setup

**Datasets and Models.** We conduct our experiment on several datasets including CIFAR-10 [32], ImageNet-1K [13] and CelebA-HQ [29]/CelebA [36]. For neural network models, we use the available models on RobustBench [11] (see more details in the next paragraph), which includes 17/5 models on CIFAR-10/ImageNet, correspondingly. We also use several off-the-shelf GANs and

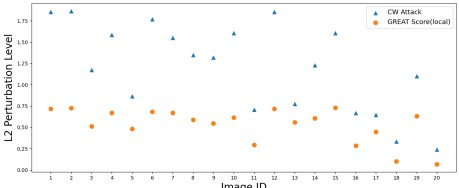
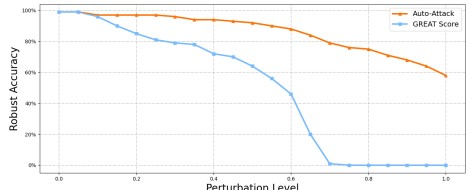

Figure 1: Comparison of local GREAT Score and CW attack in $\mathcal{L}_2$ perturbation on CIFAR-10 with Rebuffi_extra model [46]. The x-axis is the image id. The result shows the local GREAT Score is indeed a lower bound of the perturbation level found by CW attack.

Figure 2: Cumulative robust accuracy (RA) with varying $\mathcal{L}_2$ perturbation level using 500 samples. Note that GREAT Score gives a certified RA for attack-proof robustness, whereas Auto-Attack is an empirical robustness evaluation.

diffusion models (DMs) trained on CIFAR-10 and ImageNet for computing GREAT Score in an ablation study (we defer the model details to later paragraphs).

**Summary of Classifiers on RobustBench.** The RobustBench [49] is to-date the largest benchmark for robustness evaluation with publicly accessible neural network models submitted by contributors. RobustBench uses the default test dataset from several standard image classification tasks, such as CIFAR-10 and ImageNet-1K, to run Auto-Attack [10] and report the resulting accuracy with $\mathcal{L}_2$-norm and $\mathcal{L}_\infty$-norm perturbations (i.e., the robust accuracy – RA) as a metric for adversarial robustness. Even under one perturbation type, it is not easy to make a direct and fair comparison among all submitted models on RobustBench because they often differ by the training scheme, network architecture, as well as the usage of additional real and/or synthetic data. To make a meaningful comparison with GREAT Score, we select all non-trivial models (having non-zero RA) submitted to the CIFAR-10 and ImageNet-1K benchmarks and evaluated with $\mathcal{L}_2$-norm perturbation with a fixed perturbation level of 0.5 using Auto-Attack. We list the model names in Table 1 and provide their descriptions in Appendix G.

**GANs and DMs.** We used off-the-shelf GAN models provided by StudioGAN [41], a library containing released GAN models. StudioGAN also reports the Inception Score (IS) to rank the model quality. We use the GAN model with the highest IS value as our default GAN for GREAT Score, which are StyleGAN2 [30]/ BigGAN [6] for CIFAR-10 /ImageNet with IS = 10.477/99.705, respectively. For the ablation study of using different generative models in GREAT Score (Section 4.4), we also use the following GAN/DM models: LSGAN [39], GGAN [35], SAGAN [65], SNGAN [42], DDPM [27] and StyleGAN2 [30].

**GREAT Score implementation.** The implementation follows Algorithm 1 in Appendix **??** with a sigmoid/softmax function on the logits of the CIFAR-10/ImageNet classifier to ensure the model output of each dimension is within $[0, 1]$, as implied by Theorem 1. As ImageNet-1K has 1000 classes, applying sigmoid will make the robustness score function in (3) degenerate. We use softmax instead. 500 samples drawn from a generative model were used for computing GREAT Score.

**Comparative methods.** We compare the effectiveness of GREAT Score in two objectives: robustness ranking (global robustness) and per-sample perturbation. For the former, we compare the RA reported in RobustBench on the test dataset (named RobustBench Accuracy) as well as the RA of Auto-Attack on the generated data samples (named AutoAttack Accuracy). For the latter, we report the RA of Auto-Attack in $\mathcal{L}_2$-norm with a fixed perturbation level of 0.5.

**Evaluation metrics.** For robustness ranking, we report Spearman's rank correlation coefficient between two sets of model rankings (e.g., GREAT Score v.s. RobustBench Accuracy). A value closer to 1 means higher consistency. Robust accuracy refers to the fraction of correctly classified samples against adversarial perturbations.

**Calibration Method.** We run $\mathcal{L}_2$-norm CW attack [7] (with learning rate 0.005 and 200 iterations) on each generated data sample to find the minimal adversarial perturbation. Then, we use grid search in the range [0,2] with an interval of 0.00001 to find temperature value maximizing the Spearmans' rank correlation coefficient between GREAT Score and CW attack distortion.

**Compute Resources.** All our experiments were run on a GTX 2080 Ti GPU with 12GB RAM.

Table 1: Comparison of (Calibrated) GREAT Score v.s. minimal distortion found by CW attack [7] on CIFAR-10. The results are averaged over 500 samples from StyleGAN2.

| Model Name | RobustBench Accuracy(%) | AutoAttack Accuracy(%) | GREAT Score | Calibrated GREAT Score | CW Distortion |
|---|---|---|---|---|---|
| Rebuffi_extra [46] | 82.32 | 87.20 | 0.507 | 1.216 | 1.859 |
| Gowal_extra [21] | 80.53 | 85.60 | 0.534 | 1.213 | 1.324 |
| Rebuffi_70_ddpm [46] | 80.42 | 90.60 | 0.451 | 1.208 | 1.943 |
| Rebuffi_28_ddpm [46] | 78.80 | 90.00 | 0.424 | 1.214 | 1.796 |
| Augustin_WRN_extra [3] | 78.79 | 86.20 | 0.525 | 1.206 | 1.340 |
| Sehwag [54] | 77.24 | 89.20 | 0.227 | 1.143 | 1.392 |
| Augustin_WRN [3] | 76.25 | 86.40 | 0.583 | 1.206 | 1.332 |
| Rade [45] | 76.15 | 86.60 | 0.413 | 1.200 | 1.486 |
| Rebuffi_R18[46] | 75.86 | 87.60 | 0.369 | 1.210 | 1.413 |
| Gowal [21] | 74.50 | 86.40 | 0.124 | 1.116 | 1.253 |
| Sehwag_R18 [54] | 74.41 | 88.60 | 0.236 | 1.135 | 1.343 |
| Wu2020Adversarial [62] | 73.66 | 84.60 | 0.128 | 1.110 | 1.369 |
| Augustin2020Adversarial [3] | 72.91 | 85.20 | 0.569 | 1.199 | 1.285 |
| Engstrom2019Robustness [15] | 69.24 | 82.20 | 0.160 | 1.020 | 1.084 |
| Rice2020Overfitting [47] | 67.68 | 81.80 | 0.152 | 1.040 | 1.097 |
| Rony2019Decoupling [50] | 66.44 | 79.20 | 0.275 | 1.101 | 1.165 |
| Ding2020MMA [14] | 66.09 | 77.60 | 0.112 | 0.909 | 1.095 |

Table 2: Spearman's rank correlation coefficient on CIFAR-10 using GREAT Score, RobustBench (with test set), and Auto-Attack (with generated samples).

| | Uncalibrated | Calibrated |
|---|---|---|
| GREAT Score vs. RobustBench Correlation | 0.6618 | 0.8971 |
| GREAT Score vs. AutoAttack Correlation | 0.3690 | 0.6941 |
| RobustBench vs. AutoAttack Correlation | 0.7296 | 0.7296 |

## 4.2 Local and Global Robustness Analysis

Recall from Theorem 1 that the local robustness score proposed in (3) gives a certified perturbation level for generated samples from a generative model. To verify this claim, we randomly select 20 generated images on CIFAR-10 and compare their local certified perturbation level to the perturbation found by the CW attack [7] using the Rebuffi_extra model [46]. Figure 1 shows the perturbation level of local GREAT Score in (3) and that of the corresponding CW attack per sample. We can see that the local GREAT Score is a lower bound of CW attack, as the CW attack finds a successful adversarial perturbation that is no smaller than the minimal perturbation $\Delta_{\min}$ (i.e., an over-estimation). The true $\Delta_{\min}$ value lies between these lower and upper bounds.

In Figure 2, we compare the cumulative robust accuracy (RA) of GREAT Score and Auto-Attack over 500 samples by sweeping the $\mathcal{L}_2$ perturbation level from 0 to 1 with a 0.05 increment for Auto-Attack. The cumulative RA of GREAT Score at a perturbation level $r$ represents the fraction of samples with local GREAT Scores greater than $r$, providing an attack-proof guarantee that no attacks can achieve a lower RA at the same perturbation level. For Auto-Attack, the RA at each perturbation level is calculated as the fraction of correctly classified samples under that specific perturbation. The blue curve in the figure represents the RA from empirical Auto-Attack, while the orange curve shows the RA derived from GREAT Score, offering a certified robustness guarantee. We observe that the trend of attack-independent certified robustness (GREAT Score) closely mirrors that of empirical attacks (Auto-Attack), suggesting that GREAT Score effectively reflects empirical robustness. It is important to note that the gap between our certified curve and the empirical curve of AutoAttack does not necessarily indicate inferiority of GREAT Score. Instead, this discrepancy could point to the existence of undiscovered adversarial examples at higher perturbation radii. This gap illustrates the fundamental difference between certified and empirical robustness measures, highlighting the potential for GREAT Score to provide a more conservative, yet guaranteed, estimate of model robustness.

Table 1 compares the global robustness statistics of the 17 grouped CIFAR-10 models on RobustBench for uncalibrated and calibrated versions respectively, in terms of the GREAT Score and the average distortion of CW attack, which again verifies GREAT Score is a certified lower bound on the true global robustness (see its definition in Section 3.1), while any attack with 100% attack success rate only gives an upper bound on the true global robustness. We also observe that calibration can indeed enlarge the GREAT Score and tighten its gap to the distortion of CW attack.

## 4.3 Model Ranking on CIFAR-10 and ImageNet

Following the experiment setup in Section 4.1, we compare the model ranking on CIFAR-10 using GREAT Score (evaluated with generated samples), RobustBench (evaluated with Auto-Attack on the test set), and Auto-Attack (evaluated with Auto-Attack on generated samples). Table 2 presents their mutual rank correlation (higher value means more aligned ranking) with calibrated and uncalibrated versions. We note that there is an innate discrepancy between Spearman's rank correlation coefficient (way below 1) of RobustBench v.s. Auto-Attack, which means Auto-Attack will give inconsistent model rankings when evaluated on different data samples. In addition, GREAT Score measures *classification margin*, while AutoAttack measures *accuracy* under a fixed perturbation budget $\epsilon$.

AutoAttack's ranking will change if we use different $\epsilon$ values. E.g., comparing the ranking of $\epsilon = 0.3$ and $\epsilon = 0.7$ on 10000 CIFAR-10 test images for AutoAttack, the Spearman's correlation is only 0.9485. Therefore, we argue that GREAT Score and AutoAttack are *complementary* evaluation metrics and they don't need to match perfectly. Despite their discrepancy, before calibration, the correlation between GREAT Score and RobustBench yields a similar value. With calibration, there is a significant improvement in rank correlation between GREAT Score to Robustbench and Auto-Attack, respectively.

Table 3 presents the global robustness statistics of these three methods on ImageNet. We observe almost perfect ranking alignment between GREAT Score and RobustBench, with their Spearman's rank correlation coefficient being 0.8, which is higher than that of Auto-Attack and RobustBench (0.6). These results suggest that GREAT Score is a useful metric for *margin-based* robustness evaluation.

## 4.4 Ablation Study and Run-time Analysis

**Ablation study on GANs and DMs.** Evaluating on CIFAR-10, Figure 3 compares the inception score (IS) and the Spearman's rank correlation coefficient between GREAT Score and RobustBench on five GANs and DDPM. One can observe that models with higher IS attain better ranking consistency.

**Limitations and Further Analysis for generation models.** While our experiments demonstrate the effectiveness of GREAT Score, it's important to acknowledge certain limitations and provide further analysis. The performance of GREAT Score relies on the generative model's ability to produce valid samples belonging to the conditioned class. Recent studies [34, 53] have shown GANs' convergence to true data distributions under specific conditions, and our experiments further demonstrate high-quality instances produced by the generative models, as evidenced by the inception score and the strong Spearman's rank correlation between GREAT Score and RobustBench. We recognize that in some cases, class ambiguity may exist. However, given our focus on evaluating classifier robustness, we typically deal with well-defined and distinctive labels, considering the issue of label ambiguity is beyond the scope of our method. Furthermore, the assumption that the generative model provides a good approximation of the true data-generating distribution is crucial. Recent work [34, 53] has also demonstrated the convergence rate of approaching the true data distribution for a family of GANs under certain conditions. These considerations highlight areas for potential future work and underscore the importance of careful generative model selection when applying GREAT Score.

**Run-time analysis.** Figure 4 compares the run-time efficiency of GREAT Score over Auto-Attack on the same 500 generated CIFAR-10 images. We show the ratio of their average per-sample run-time (wall clock time of GREAT Score/Auto-Attack is reported in Appendix I) and observe around 800-2000 times improvement, validating the computational efficiency of GREAT Score. Furthermore, our framework demonstrates excellent scalability with increasing dataset sizes and model complexity, as detailed in Appendix N, showing linear scaling behavior that makes it suitable for large-scale applications.

**Sample Complexity and GREAT Score.** In Appendix J, we report the mean and variance of GREAT Score with a varying number of generated data samples. The results show that the statistics of GREAT Score are quite stable even with a small number of data samples (i.e., $\geq$500).

Table 3: Robustness evaluation on ImageNet using GREAT Score, RobustBench (with test set), and Auto Attack (with generated samples). The Spearman's rank correlation coefficient for GREAT Score v.s. RobustBench and Auto-Attack v.s. RobustBench is 0.9 and 0.872, respectively.

| Model Name | RobustBench Accuracy (%) | AutoAttack Accuracy (%) | GREAT Score |
|---|---|---|---|
| Trans1 [52] | 38.14 | 30.4 | 0.504 |
| Trans2 [52] | 34.96 | 25.8 | 0.443 |
| LIBRARY [15] | 29.22 | 30.6 | 0.449 |
| Fast [61] | 26.24 | 19.2 | 0.273 |
| Trans3 [52] | 25.32 | 19.6 | 0.275 |

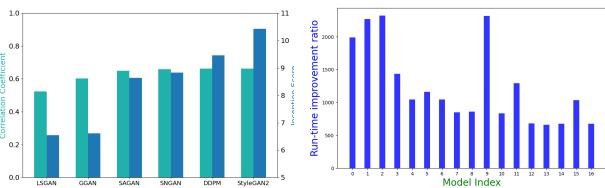

Figure 3: Comparison of Inception Score and Spearman's rank correlation to RobustBench using GREAT Score with different GANs.

Figure 4: Run-time improvement (GREAT Score over Auto-Attack) on 500 generated CIFAR-10 images.

Table 4: Group-wise and overall robustness evaluation for online gender classification APIs over 500 generated samples (per group).

| Online API Name | Old | Young | With Eyeglasses | Without Eyeglasses | Total |
|---|---|---|---|---|---|
| BetaFace | 0.950 | 0.662 | 0.547 | 0.973 | 0.783 |
| Inferdo | 0.707 | 0.487 | 0.458 | 0.669 | 0.580 |
| ARSA-Technology | 1.033 | 0.958 | 0.739 | 1.082 | 0.953 |
| DEEPFACE | 0.979 | 0.774 | 0.763 | 0.969 | 0.872 |
| Baidu | 1.097 | 1.029 | 0.931 | 1.134 | 1.048 |
| Luxand | 1.091 | 0.912 | 0.673 | 1.010 | 0.944 |

Table 5: GREAT Score v.s. robust accuracy under square attack [1].

| DEEPFACE | Old | Young | With Eyeglasses | Without Eyeglasses |
|---|---|---|---|---|
| Square Attack | 84.40% | 72.60% | 65.80% | 89.00% |
| GREAT Score | 0.979 | 0.774 | 0.763 | 0.969 |

## 4.5 Evaluation on Online Facial Recognition APIs

To demonstrate GREAT Score enables robustness evaluation of black-box models that only provide model inference outcomes based on date inputs, we use synthetically generated face images with hidden attributes to evaluate six online face recognition APIs for gender classification. It is worth noting that GREAT Score is suited for privacy-sensitive assessment because it only uses synthetic face images for evaluation and does not require using real face images.

We use an off-the-shelf face image generator InterFaceGAN [56] trained on CelebA-HQ dataset [29], which can generate controllable high-quality face images with the choice of attributions such as eyeglasses, age, and expression. We generate four different groups (attributes) of face images for evaluation: Old, Young, With Eyeglasses, and Without Eyeglasses. For annotating the ground truth gender labels of the generated images, we use the gender predictions from the FAN classifier [25]. In total, 500 gender-labeled face images are generated for each group. Appendix L shows some examples of the generated images for each group.

We evaluate the GREAT Score on six online APIs for gender classification: BetaFace [5], Inferdo [28], Arsa-Technology [2], DeepFace [55], Baidu [4] and Luxand [37]. These APIs are "black-box" models to end users or an external model auditor because the model details are not revealed and only the model inference results returned by APIs (prediction probabilities on Male/Female) are provided.

Finally, we upload these images to the aforementioned online APIs and calculate the GREAT Score using the returned prediction results. Table 4 displays the group-level and overall GREAT Score results. Our evaluation reveals interesting observations. For instance, APIs such as BetaFace, Inferno, and DEEPFACE exhibit a large discrepancy for Old v.s. Young, while other APIs have comparable scores. For all APIs, the score of With Eyeglasses is consistently and significantly lower than that of Without Eyeglasses, which suggests that eyeglasses could be a common spurious feature that affects the group-level robustness in gender classification. The analysis demonstrates how GREAT Score can be used to study the group-level robustness of an access-limited model in a privacy-enhanced manner.

To verify our evaluation, in Table 5 we compare GREAT Score to the black-box square attack [1] with $\epsilon = 2$ and # queries$= 100$ on DEEPFACE. For both Age and Eyeglasses groups (Old v.s. Young and W/ v.s. W/O eyeglasses), we see consistently that a higher GREAT Score (second row) indicates better robust accuracy (%, first row) against square attack.

## 5 Conclusion

In this paper, we presented GREAT Score, a novel and computation-efficient attack-independent metric for global robustness evaluation against adversarial perturbations. GREAT Score uses an off-the-shelf generative model such as GANs for evaluation and enjoys theoretical guarantees on its estimation of the true global robustness. Its computation is lightweight and scalable because it only requires accessing the model predictions on the generated data samples. Our extensive experimental results on CIFAR-10 and ImageNet also verified high consistency between GREAT Score and the attack-based model ranking on RobustBench, demonstrating that GREAT Score can be used as an efficient measure complementary to existing robustness benchmarks. We also demonstrated the novel use of GREAT Score for the robustness evaluation of online facial recognition APIs.

**Limitations.** One limitation could be that our framework of global adversarial robustness evaluation using generative models is centered on $\mathcal{L}_2$-norm based perturbations. This limitation could be addressed if the Stein's Lemma can be extended for other $\mathcal{L}_p$ norms.

## Acknowledgments and Disclosure of Funding

This work was supported by the JC STEM Lab of Intelligent Design Automation funded by The Hong Kong Jockey Club Charities Trust for Zaitang Li and Tsung-Yi Ho. Also, this material is based upon work supported by the Chief Digital and Artificial Intelligence Office under Contract No. W519TC-23-9-2037 for Pin-Yu Chen.

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

# A  Notations

Table 6: Main notations used in this paper

| Notation | Description |
|---|---|
| $d$ | dimensionality of the input vector |
| $K$ | number of output classes |
| $f : \mathbb{R}^d \to \mathbb{R}^K$ | neural network classifier |
| $x \in \mathbb{R}^d$ | data sample |
| $y$ | groundtruth class label |
| $\delta \in \mathbb{R}^d$ | input perturbation |
| $\|\delta\|_p$ | $\mathcal{L}_p$ norm of perturbation, $p \geq 1$ |
| $\Delta_{\min}$ | minimum adversarial perturbation |
| $G$ | (conditional) generative model |
| $z \sim \mathcal{N}(0, I)$ | latent vector sampled from Gaussian distribution |
| $g$ | robustness score function defined in (3) |
| $\Omega(f)/\widehat{\Omega}(f)$ | true/estimated global robustness defined in Section 3.1 |

# B  More Motivations on using Generative Models for Robustness Evaluation

We emphasize the necessity of generative models using the points below.

1. *Global robustness assessment requires a GM.* The major focus and novelty of our study are to evaluate the global robustness with respect to the underlying true data distribution, and we propose to use a GM as a proxy. We argue that such a proxy is necessary to evaluate global robustness unless the true data distribution is known.

2. *GAN can provably match data distribution.* Recent works such as [53] and [34] have proved the convergence rate of approaching the true data distribution for a family of GANs under certain conditions. This will benefit global robustness evaluation (see Figure 3 for ablations on GAN variants).

3. *Privacy-sensitive remote model auditing.* As shown in Sec 4.5, synthetic data from generative models can facilitate the robustness evaluation of privacy-sensitive models.

## B.1  Related Works for Global Robustness Evaluation for Deep Neural Networks.

There are some works studying "global robustness", while their contexts and scopes are different than ours. In [51], the global robustness is defined as the expectation of the maximal certified radius of $\mathcal{L}_0$-norm over a test dataset. Ours is not limited to a test set, and we take the novel perspective of the entire data distribution and use a generative model to define and evaluate global robustness. The other line of works considers deriving and computing the global Lipschitz constant of the classifier as a global certificate of robustness guarantee, as it quantifies the maximal change of the classifier with respect to the entire input space [33]. The computation can be converted as a semidefinite program (SDP) [16]. However, the computation of SDP is expensive and hard to scale to larger neural networks. Our method does not require computing the global Lipschitz constant, and our computation is as simple as data forward pass for model inference.

# C  Proof of Theorem 1

In this section, we will give detailed proof for the certified global robustness estimate in Theorem 1. The proof contains three parts: (i) derive the local robustness certificate; (ii) derive the closed-form global Lipschitz constant; and (iii) prove the proposed global robustness estimate is a lower bound on the true global robustness.

We provide a proof sketch below:

1. We use the local robustness certificate developed in [59], which shows an expression of a certified (attack-proof) $\mathcal{L}_p$-norm bounded perturbation for any $p \geq 1$. The certificate is a function of the gap between the best and second-best class predictions, as well as a local Lipschitz constant associated with the gap function.

2. We use Stein's Lemma [57] which states that the mean of a measurable function integrated over a zero-mean isotropic Gaussian distribution has a closed-form global Lipschitz constant in the $\mathcal{L}_2$-norm. This result helps avoid the computation of the local Lipschitz constant in Step 1 for global robustness evaluation using generative models.

3. We use the results from Steps 1 and 2 to prove that the proposed global robustness estimate $\widehat{\Omega}(f)$ is a lower bound on the true global robustness $\Omega(f)$ with respect to $G$.

### C.1 Local robustness certificate

In this part, we use the local robustness certificate in [59] to show an expression for local robustness certificate consisting of a gap function in model output and a local Lipschitz constant. The first lemma formally defines Lipschitz continuity and the second lemme introduces the the local robustness certificate in [59].

**Lemma 1** (Lipschitz continuity in Gradient Form ([44])). *Let $S \subset \mathbf{R}^d$ be a convex bound closed set and let $f : S \to \mathbf{R}$ be a continuously differentiable function on an open set containing $S$. Then $f$ is a Lipschitz continuous function if the following inequality holds for any $x, y \in S$ :*

$$|f(x) - f(y)| \le L_q \|x - y\|_p \tag{5}$$

*where $L_q = \max_{x \in S} \|\nabla f(x)\|_q :$ is the corresponding Lipschitz constant, and $\nabla f(x) = (\frac{\partial f}{\partial x_1}, \dots \frac{\partial f}{\partial x_d})^\top$ is the gradient of the function f(x), and $1/q + 1/p = 1$, $p \ge 1, q \le \infty$.*

We say $f$ is $L_q$-continuous in $\mathcal{L}_p$ norm if (5) is satisfied.

**Lemma 2** (Formal guarantee on lower bound for untargeted attack of Theorem 3.2 in [59]). *Let $x_0 \in \mathbf{R}^d$ and $f : \mathbf{R}^d \to \mathbf{R}^K$ be a multi-class classifier, and $f_i$ be the $i$-th output of $f$. For untargeted attack, to ensure that the adversarial examples can not be found for each class, for all $\delta \in \mathbf{R}^d$, the lower bound of minumum distortion can be expressed by:*

$$\|\delta\|_p \le \min_{i \ne m} \frac{f_m(x_0) - f_i(x_0)}{L_q^i} \tag{6}$$

*where $m = \arg\max_{i \in \{1, \dots, K\}} f_i(x_0)$, $1/q + 1/p = 1$, $p \ge 1, q \le \infty$, and $L_q^i$ is the Lipschitz constant for the function $f_m(x) - f_i(x)$ in $L_q$ norm.*

### C.2 Proof of closed-form global Lipschitz constant in the $L_2$-norm over Gaussian distribution

In this part, we present two lemmas towards developing the global Lipschitz constant of a function smoothed by a Gaussian distribution.

**Lemma 3** (Stein's lemma [57]). *Given a soft classifier $F : \mathbf{R}^d \to \mathbf{P}$, where $\mathbf{P}$ is the space of probability distributions over classes. The associated smooth classifier with parameter $\sigma \ge 0$ is defined as:*

$$\bar{F} := (F * \mathcal{N}(0, \sigma^2 I))(x) = \mathbb{E}_{\delta \sim \mathcal{N}(0, \sigma^2 I)}[F(x + \delta)] \tag{7}$$

*Then, $\bar{F}$ is differentiable, and moreover,*

$$\nabla \bar{F} = \frac{1}{\sigma^2} \mathbb{E}_{\delta \sim \mathcal{N}(0, \sigma^2 I)}[\delta \cdot F(x + \delta)] \tag{8}$$

In a lecture note[1], Li used Stein's Lemma [57] to prove the following lemma:

**Lemma 4** (Proof of global Lipschitz constant). *Let $\sigma \ge 0$, let $h : \mathbb{R}^d \to [0, 1]$ be measurable, and let $H = h * \mathcal{N}(0, \sigma^2 I)$. Then $H$ is $\sqrt{\frac{2}{\pi\sigma^2}}$ – continuous in $L_2$ norm*

---

[1]https://jerryzli.github.io/robust-ml-fall19/lec14.pdf

## C.3 Proof of the proposed global robustness estimate $\widehat{\Omega}(f)$ is a lower bound on the true global robustness $\Omega(f)$ with respect to $G$

Recall that we assume a generative model $G(\cdot)$ generates a sample $G(z)$ with $z \sim \mathcal{N}(0, I)$. Following the form of Lemma 2 (but ignoring the local Lipschitz constant), let

$$g'(G(z)) = \max\{f_c(G(z)) - \max_{k \in \{1,...,K\}, k \neq c} f_k(G(z)), 0\} \tag{9}$$

denote the gap in the model likelihood of the correct class $c$ and the most likely class other than $c$ of a given classifier $f$, where the gap is defined to be $0$ if the model makes an incorrect top-1 class prediction on $G(z)$. Then, using Lemma 4 with $g'$, we define

$$\mathbb{E}_{z \sim \mathcal{N}(0,I)}[g'(G(z))] = (g' \circ G) * \mathcal{N}(0, I) \tag{10}$$

[ZAITANG: Modified the equations] and thus $\mathbb{E}_{z \sim \mathcal{N}(0,I)}[g'(G(z))]$ has a Lipschitz constant $\sqrt{\frac{2}{\pi}}$ in $\mathcal{L}_2$ norm. This implies that for any input perturbation $\delta$,

$$|\mathbb{E}_{z \sim \mathcal{N}(0,I)}[g'(G(z) + \delta)] - \mathbb{E}_{z \sim \mathcal{N}(0,I)}[g'(G(z))]| \tag{11}$$

$$\leq \sqrt{\frac{2}{\pi}} \cdot \|\delta\|_2 \tag{12}$$

and therefore

$$\mathbb{E}_{z \sim \mathcal{N}(0,I)}[g'(G(z) + \delta)] \tag{13}$$

$$\geq \mathbb{E}_{z \sim \mathcal{N}(0,I)}[g'(G(z))] - \sqrt{\frac{2}{\pi}} \cdot \|\delta\|_2 \tag{14}$$

Note that if the right-hand side of (13) is greater than zero, this will imply the classifier attains a nontrivial positive mean gap with respect to the generative model. This condition holds for any $\delta$ satisfying $\|\delta\|_2 < \sqrt{\frac{\pi}{2}} \cdot \mathbb{E}_{z \sim \mathcal{N}(0,I)}[g'(G(z))]$. Note that by definition any minimum perturbation on $G(z)$ will be no smaller than $\sqrt{\frac{\pi}{2}} \cdot \mathbb{E}_{z \sim \mathcal{N}(0,I)}[g'(G(z))]$ as it will make $g'(G(z)) = 0$ almost surely. Therefore, by defining $g = \sqrt{\frac{\pi}{2}} \cdot g'$, we conclude that the global robustness estimate $\widehat{\Omega}(f)$ in (2) using the proposed local robustness score $g$ defined in (3) is a certified lower bound on the true global robustness $\Omega(f)$ with respect to $G$.

# D   Proof of Theorem 2

To prove Theorem 2, we first define some notations as follows, with a slight abuse of the notation $f$ as a generic function in this part. For a vector of independent random variables $X = (X_1..., X_n)$, define $X' = (X'_1..., X'_n)$ to be i.i.d. to X, $x = (x_1, ..., x_n) \in \mathbf{X}$, and the sub-exponential norms $\|\cdot\|_{\psi_2}$ for any random variable $Z$ as

$$\|Z\|_{\psi_2} = \sup_{p \geq 1} \frac{\|Z\|_p}{\sqrt{p}} \tag{15}$$

Let $f : X^n \mapsto \mathbf{R}$. We further define the $k$-th centered conditional version of $f$ as :

$$f_k(X) = f(X) - \mathbb{E}[f(X)|X_1, ..., X_{k-1}, X_{k+1}, ...X_n] \tag{16}$$

**Lemma 5** (Concentration inequality from Theorem 3.1 in [40]). *Let $f : X^n \mapsto \mathbf{R}$ and $X = (X_1, \ldots, X_n)$ be a vector of independent random variables with values in a space $\mathbb{X}$. Then for any $t > 0$ we have*

$$Pr(f(X) - E[f(X')] > t) \leq \exp\left(\frac{-t^2}{32e\left\|\sum_k \|f_k(X)\|_{\psi_2}^2\right\|_\infty}\right) \tag{17}$$

Recall that we aim to derive a probabilistic guarantee on the sample mean of the local robustness score in (3) from a $K$-way classifier with its outputs bounded by $[0,1]^K$. Following the definition of $g$ (for simplicity, ignoring the constant $\sqrt{\pi/2}$), the sample mean $f$ can be expressed as:

$$f(X) = \frac{1}{n}\sum_{i=1}^{n} g(X_i) \tag{18}$$

where $X_i \sim \mathcal{N}(0, I)$.

Following the definition of (16),

$$f_k(X) = f(X) - \mathbb{E}[f(X)|X_1, ..., X_{k-1}, X_{k+1}, ...X_n] \tag{19}$$

$$= \frac{1}{n}[g(X_k) - g(X_k')] \leq \frac{1}{n} \tag{20}$$

This implies $f_k(X)$ is bounded by $\frac{1}{n}$, i.e., $\|f_k(X)\|_\infty \leq \frac{1}{n}$, and also $\|f_k(X)\|_{\psi_2} \leq \frac{1}{n}$.

Squaring over $\|f_k(X)\|_{\psi_2}$ gives

$$\|f_k(X)\|_{\psi_2}^2 \leq \frac{1}{n^2} \tag{21}$$

As a result,

$$\left\|\sum_k \|f_k(X)\|_{\psi_2}^2\right\|_\infty \leq n \cdot \frac{1}{n^2} = \frac{1}{n} \tag{22}$$

Divide both side of (22) and multiply with $\dfrac{-t^2}{32e}$ gives:

$$\frac{-t^2}{32e\left\|\sum_k \|f_k(X)\|_{\psi_2}^2\right\|_\infty} \leq \frac{-t^2 n}{32e} \tag{23}$$

Take exponential function over both side of (23) gives

$$\exp\left(\frac{-t^2}{32e\left\|\sum_k \|f_k(X)\|_{\psi_2}^2\right\|_\infty}\right) \leq \exp\left(\frac{-t^2 n}{32e}\right) \tag{24}$$

Recall Lemma 5, since this bound holds on both sides of the central mean, we rewrite it as:

$$\text{Prob}(|f(X) - \mathbb{E}[f(X')]| > t) \leq 2\exp\left(\frac{-t^2}{32e\left\|\sum_k \|f_k(X)\|_{\psi_2}^2\right\|_\infty}\right) \tag{25}$$

Hence to ensure that given a statistical tolerance $\epsilon > 0$ with $\delta$ as the maximum outage probability, i.e., $\text{Prob}(|f(X) - E[f(X')]| > \epsilon) \leq \delta$, we have

$$2 \cdot \exp\left(\frac{-\epsilon^2}{32e\left\|\sum_k \|f_k(X)\|_{\psi_2}^2\right\|_\infty}\right) \leq 2\exp\left(\frac{-\epsilon^2 n}{32e}\right) \tag{26}$$

$$\leq \delta \tag{27}$$

Finally, (26) implies that the sample complexity to reach the $(\epsilon, \delta)$ condition is $n \geq \frac{32e \cdot \log(2/\delta)}{\epsilon^2}$.

Figure 5 shows the flow chart of Algorithm 1.

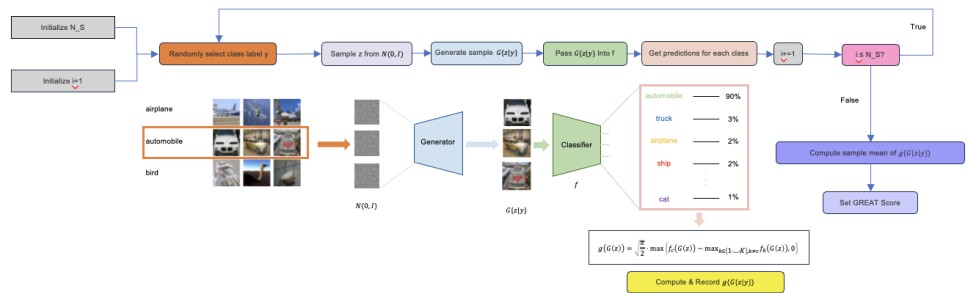

Figure 5: The Flow Chart of GREAT Score.

Table 7: Spearman's rank correlation coeffienct on CIFAR-10 using GREAT Score, RobustBench (with test set), and Auto-Attack (with generated samples) with different calibration methods.

| | GREAT Score v.s. RobustBench Correlation | GREAT Score v.s. AutoAttack Correlation | RobustBench v.s AutoAttack Correlation |
|---|---|---|---|
| softmax with temperature | -0.5024 | -0.5334 | 0.7296 |
| sigmoid with temperature | 0.7083 | 0.3641 | 0.7296 |
| sigmoid with temperature after softmax | -0.2525 | -0.2722 | 0.7296 |
| softmax with temperature after sigmoid | 0.8971 | 0.6941 | 0.7296 |

# E    Comparison between CW Attack and GREAT Score

We provide a detailed comparison of the time complexity between GREAT Score and CW Attack.

The time complexity of the GREAT Score algorithm is determined by the number of iterations (generated samples) in the loop, which is denoted as $N_S$. Within each iteration, the algorithm performs operations such as random selection, sampling from a Gaussian distribution, generating samples, and predicting class labels using the classifier. We assume these operations have constant time complexity $I$ and absorb them in the big $O$ notation. Additionally, the algorithm computes the sample mean of the recorded statistics, which involves summing and dividing the values. As there are $N_S$ values to sum and divide, this step has a time complexity of $O(N_S)$. Therefore, the overall time complexity of the algorithm can be approximated as $O(N_S \cdot I)$.

Using our nation, consider a $K$-way classifier $f$. Let $x$ be a data sample and $y$ be its top-1 classification label. Denote $\delta$ as the adversarial perturbation. The untargeted CW Attack ($L_2$ norm) solves the following optimization objective:

$$\delta^* = \arg\min_\delta(\|\delta\|_2^2 + \alpha \cdot \max\{f_y(x+\delta) - \max_{k\in\{1,...,K\},k\neq y} f_k(x+\delta),0\}) \tag{28}$$

where $f_k(\cdot)$ is the prediction of the $k$-th class, and $\alpha > 0$ is a hyperparameter.

For CW attack, the optimization process iteratively finds the adversarial perturbation. The number of iterations required depends on factors such as the desired level of attack success and the convergence criteria. Each iteration involves computing gradients, updating variables, and evaluating the objective function. It also involves a hyperparameter$\alpha$ search stage to adjust the weighted loss function.

Specifically, let $B$ be the complexity of backpropagation, $T_g$ be the number of iterative optimizations, and $T_b$ be the number of binary search steps for $\alpha$. The dominant computation complexity of CW attack for $N_S$ samples is in the order of $O(N_S \cdot T_g \cdot T_b \cdot B)$. Normally, $T_g$ is set to 1000, and $T_b$ is set to 9. Therefore, CW attack algorithm is much more time-consuming than GREAT Score.

# F    Best Calibration Coefficient on different activation methods

Table 7 shows the best ranking coefficient we achieved on each calibration option for CIFAR-10. Among all these four calibration choices, we found that Sigmoid then Temperature Softmax achieves the best result.

# G Detailed descriptions of the Models

We provide the detail description for classifiers on RobustBench in what follows. The classfiers for CIFAR-10 are mentioned first and the last paragraph provides descriptions for ImageNet classifiers.
• *Rebuffi et al. [46]:* Rebuffi et al. [46] proposed a fixing data augmentation method such as using CutMix [63] and GANs to prevent over-fitting. There are 4 models recorded in [46]: Rebuffi_extra uses extra data from Tiny ImageNet in training, while Rebuffi_70_ddpm uses synthetic data from DDPM. Rebuffi_70_ddpm/Rebuffi_28_ddpm/Rebuffi_R18 varies in the network architecture. They use WideResNet-70-16 [64]/WideResNet-28-10 [64]/PreActResNet-18 [24].
• *Gowal et al.[21]:* Gowal et al. [21] studied various training settings such as training losses, model sizes, and model weight averaging. Gowal_extra differs from Gowal in using extra data from Tiny ImageNet for training.
• *Augustin et al.[3]:* Augustin et al.[3] proposed RATIO, which trains with an out-Of-distribution dataset. Augustin_WRN_extra uses the out-of-distribution data samples for training while Augustin_WRN does not.
• *SehWag et al. [54]:* SehWag et al. [54] found that a proxy distribution containing extra data can help to improve the robust accuracy. Sehwag/Sehwag_R18 uses WideResNet-34-10 [64]/ResNet-18 [23], respectively.
• *Rade et al. [45]:* Rade [45] incorporates wrongly labeled data samples for training.
• *Wu et al. [62]:* Wu2020Adversarial [62] regularizes weight loss landscape.
• *LIBRARY:* Engstrom2019Robustness [2] is a package used to train and evaluate the robustness of neural network.
• *Rice et al. [47]:* Rice2020Overfitting [47] uses early stopping in reduce over-fitting during training.
• *Rony et al. [50]:* Rony2019Decoupling [50] generates gradient-based attacks for robust training.
• *Ding et al. [50]:* Ding2020MMA [14] enables adaptive selection of perturbation level during training.

For the 5 ImageNet models, Trans [52] incorporates transfer learning with adversarial training. Its model variants Trans1/Trans2/Trans3 use WideResNet-50-2 [64]/ResNet-50 [23]/ResNet-18 [23]. LIBRARY means using the package mentioned in Group of other models to train on ImageNet. Fast [61] means fast adversarial training. There is no $\mathcal{L}_2$-norm benchmark for ImageNet on RobustBench, so we use the $\mathcal{L}_\infty$-norm benchmark.

# H Approximation Error and Sample Complexity

Figure 6 presents the sample complexity as analyzed in Theorem 2 with varying approximation error ($\epsilon$) and three confidence parameters ($\delta$) for quantifying the difference between the sample mean and the true mean for global robustness estimation. As expected, smaller $\delta$ or smaller $\epsilon$ will lead to higher sample complexity.

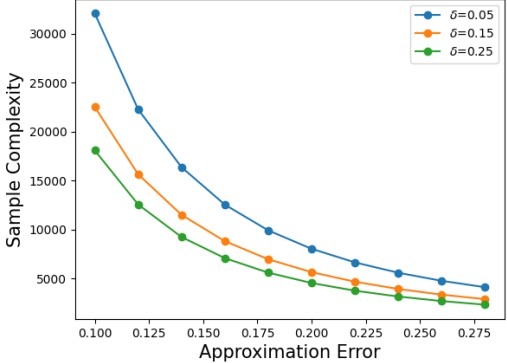

Figure 6: The relationship between the approximation error ($\epsilon$) and sample complexity in Theorem 2, with three different confidence levels: $\delta = \{5, 15, 25\}\%$.

---

[2] https://github.com/MadryLab/robustness

Table 8: Group-wise time efficiency evaluation on CIFAR-10 using GREAT Score and Auto-Attack (with 500 generated samples).

| Model Name | GREAT Score(Per Sample)(s) | AutoAttack(Per Sample)(s) |
|---|---|---|
| Rebuffi_extra [46] | 0.038 | 60.872 |
| Gowal_extra [21] | 0.034 | 59.586 |
| Rebuffi_70_ddpm [46] | 0.034 | 61.3362 |
| Rebuffi_28_ddpm [46] | 0.011 | 10.3828 |
| Augustin_WRN_extra [3] | 0.013 | 10.096 |
| Sehwag [54] | 0.011 | 10.3662 |
| Augustin_WRN [3] | 0.011 | 10.1056 |
| Rade [45] | 0.007 | 4.4114 |
| Rebuffi_R18[46] | 0.008 | 4.4644 |
| Gowal [21] | 0.034 | 60.746 |
| Sehwag_R18 [54] | 0.007 | 3.8652 |
| Wu2020Adversarial [62] | 0.012 | 10.9826 |
| Augustin2020Adversarial [3] | 0.014 | 6.9148 |
| Engstrom2019Robustness [15] | 0.012 | 6.6462 |
| Rice2020Overfitting [47] | 0.007 | 3.5776 |
| Rony2019Decoupling [50] | 0.010 | 8.5834 |
| Ding2020MMA [14] | 0.008 | 3.6194 |

# I   Complete Run-time Results

The complete run-time results of Figure 4 are given in Table 8:

# J   Sample Complexity and GREAT Score

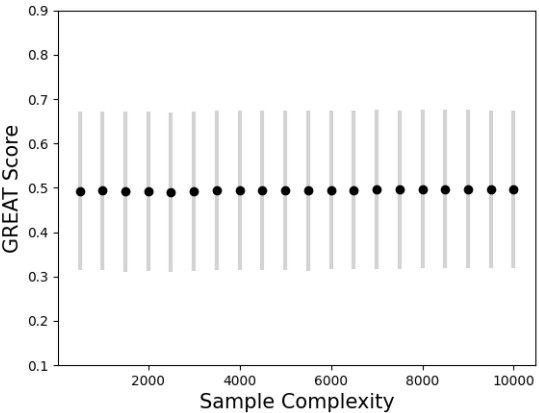

Figure 7: The relation of GREAT Score and sample complexity using CIFAR-10 and Rebuffi_extra model over (500-10000) range. The data points refer to the mean value for GREAT Score, and the error bars refers to the standard derivation for GREAT Score.

Figure 7 reports the mean and variance of GREAT Score with a varying number of generated data samples using CIFAR-10 and the Rebuffi_extra model, ranging from 500 to 10000 with 500 increment. Figure 8 reports the mean and variance of GREAT Score ranging from 50 to 1000 with 50 increment. The results show that the statistics of GREAT Score are quite stable even with a small number of data samples.

# K   GREAT Score Evaluation on the Original Test Samples of CIFAR-10

Besides evaluating the GREAT Score on the generated samples from GAN, we also run the evaluation process on 500 test samples of CIFAR-10. Table 9 shows the evaluated GREAT Score.

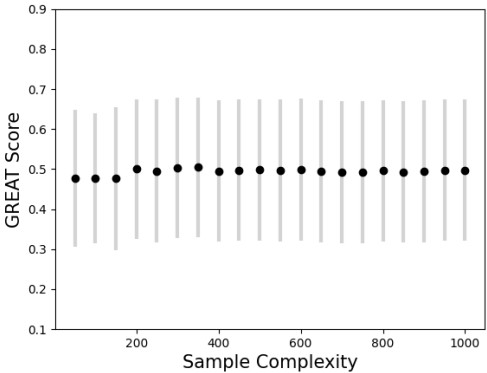

Figure 8: The relation of GREAT Score and sample complexity using CIFAR-10 and Rebuffi_extra model over (50-1000) range. The data points refer to the mean value for GREAT Score, and the error bars refers to the standard derivation for GREAT Score.

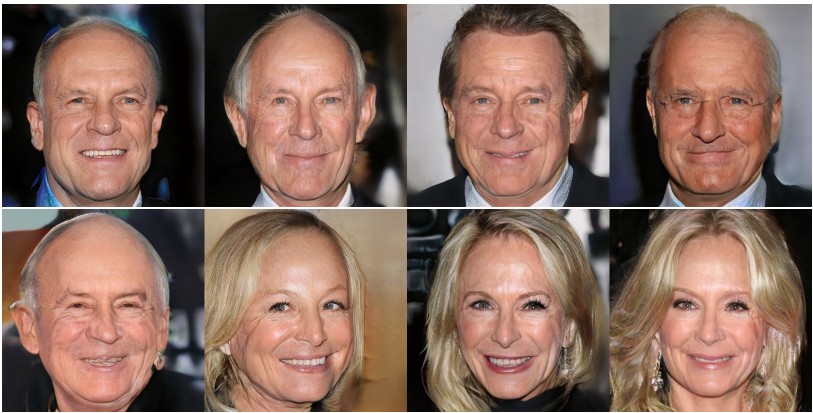

Figure 9: Generated Images for old subgroup.

## L   Generated Images from Facial GAN Models

We show the generated images from four groups in what follows.

## M   Impact Statements

As this work focuses on quantifying and scoring the global robustness of neural network classifiers, we do not currently foresee any negative impact based on our work. We envision our work to be used in model auditing settings such as model cards.

## N   Scalability Analysis

To evaluate the scalability of the GREAT Score framework, we conducted experiments using three ResNet variants (ResNet50, ResNet101, and ResNet152) with varying dataset sizes ranging from 500 to 2000 images. The computation times were measured in milliseconds without implementing any attack mechanisms.

Table 10 presents the detailed computational performance across different configurations.

Our experimental results demonstrate a linear increase in computation time with respect to both dataset size and model complexity. More sophisticated architectures like ResNet152 required proportionally more processing time compared to simpler ones like ResNet50. This linear scalability indicates that

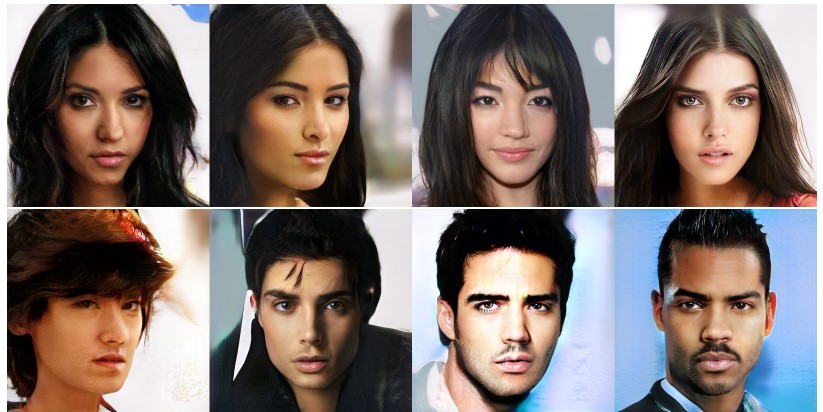

Figure 10: Generated Images for young subgroup.

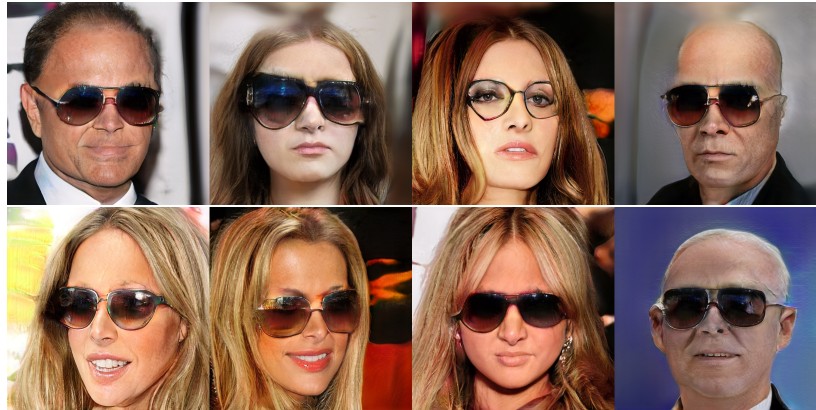

Figure 11: Generated Images for with-eyeglasses subgroup.

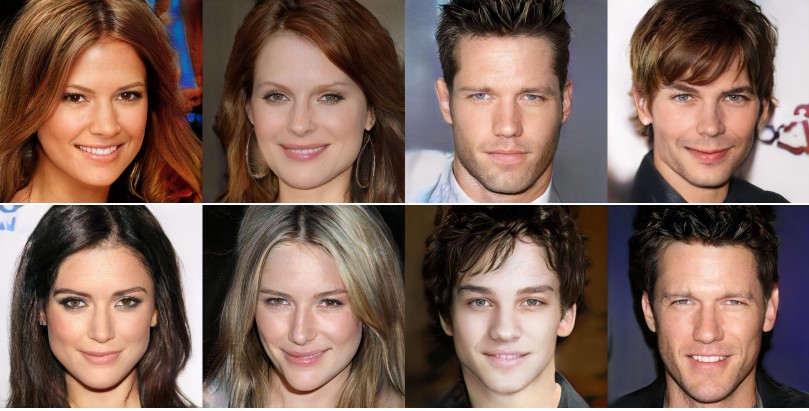

Figure 12: Generated Images for without-eyeglasses subgroup.

the GREAT Score framework efficiently handles larger datasets and more complex models, making it suitable for large-scale applications.

Table 9: GREAT Score on CIFAR-10. The results are averaged over 500 original test samples.

| Model Name | RobustBench Accuracy(%) | AutoAttack Accuracy(%) | GREAT Score | Test Samples GREAT Score |
|---|---|---|---|---|
| Rebuffi_extra [46] | 82.32 | 87.20 | 0.507 | 0.465 |
| Gowal_extra [21] | 80.53 | 85.60 | 0.534 | 0.481 |
| Rebuffi_70_ddpm [46] | 80.42 | 90.60 | 0.451 | 0.377 |
| Rebuffi_28_ddpm [46] | 78.80 | 90.00 | 0.424 | 0.344 |
| Augustin_WRN_extra [3] | 78.79 | 86.20 | 0.525 | 0.525 |
| Sehwag [54] | 77.24 | 89.20 | 0.227 | 0.227 |
| Augustin_WRN [3] | 76.25 | 86.40 | 0.583 | 0.489 |
| Rade [45] | 76.15 | 86.60 | 0.413 | 0.331 |
| Rebuffi_R18[46] | 75.86 | 87.60 | 0.369 | 0.297 |
| Gowal [21] | 74.50 | 86.40 | 0.124 | 0.109 |
| Sehwag_R18 [54] | 74.41 | 88.60 | 0.236 | 0.176 |
| Wu2020Adversarial [62] | 73.66 | 84.60 | 0.128 | 0.106 |
| Augustin2020Adversarial [3] | 72.91 | 85.20 | 0.569 | 0.493 |
| Engstrom2019Robustness [15] | 69.24 | 82.20 | 0.160 | 0.127 |
| Rice2020Overfitting [47] | 67.68 | 81.80 | 0.152 | 0.120 |
| Rony2019Decoupling [50] | 66.44 | 79.20 | 0.275 | 0.221 |
| Ding2020MMA [14] | 66.09 | 77.60 | 0.112 | 0.08 |

Table 10: Computation time (ms) for different ResNet models and dataset sizes

| Dataset Size | ResNet50 | ResNet101 | ResNet152 |
|:---:|:---:|:---:|:---:|
| 500 | 3274 | 6251 | 9149 |
| 1000 | 6529 | 12528 | 18339 |
| 1500 | 9785 | 18838 | 27481 |
| 2000 | 12960 | 24917 | 36588 |

