# OpenReview forum: "GREAT Score: Global Robustness Evaluation of Adversarial Perturbation using Generative Models"
_NeurIPS.cc/2024/Conference — NeurIPS 2024 poster_

### Official Review · Reviewer_P6PA · 2024-06-12

**Soundness:** 3
**Presentation:** 2
**Contribution:** 4
**Rating:** 7
**Confidence:** 5

**Summary:**

This paper presents an innovative adversarial robustness measure, which leverages a generative model to produce data samples, record the marginal confidence score as a local statistic and average them over the data distribution. The proposed measure is designed to be efficient, scalable, and potentially applicable to unknown data distributions. Empirical validation is conducted using local models and commercial inference APIs, demonstrating the utility of the robustness evaluation.

The concept introduced in this study is commendable for its originality, and the metric indeed offers valuable insights into model robustness. Nonetheless, it requires substantial revisions to enhance its clarity, presentation, and justification of claims before it can be accepted.

**Strengths:**

1. The metric introduced is a pioneering approach for assessing model robustness, characterized by its attack-independence, scalability, and potential applicability to unknown data distributions.
2. A theoretical analysis is provided, establishing that the metric serves as a lower bound for the probabilistic minimum adversarial perturbation.
3. The practicality of the proposed measure is supported by experimental validation on commercial black-box APIs.
4. There is a demonstrated strong correlation between the proposed metric and robust accuracy, suggesting the metric's effectiveness.

**Weaknesses:**

1. Presentation issues that may lead to confusion include:
   (1) The second paragraph of introduction lacks a precise definition of adversarial robustness evaluation, which could be problematic for less experienced readers.
   (2) Putting the testing algorithm in the appendix hurts the coherence of the paper. It would be better to include it in the main text.
   (3) Figure 2 requires additional clarification to elucidate how robust accuracy (RA) and the proposed metric are integrated into the same plot. Current discussion is insufficient.

2. The generative model's training requires at least partial knowledge of the data distribution. So the claim that the proposed metric can scale to unknown data distribution needs justification.

3. The metric's performance is contingent on the generative model's capacity to produce benign samples, yet no guarantee is provided that the generative model's ability to do so.

4. The endeavor to train a generative model to produce benign samples should be considered in making it a cost-effective and scalable solution. Maybe this metric can consider online learning to update the generative model. Hope a discussion can be provided on this.

5. The claim regarding the limitation to white-box settings (Page 2, Line 56) is inaccurate, as adversarial accuracy can also be assessed in black-box scenarios, evidenced by the effectiveness of the Square Attack method.

**Questions:**

1. The paper omits discussion of several significant works that evaluate model robustness from different perspectives. The authors should consider addressing the following studies in the related works section:
   [1] "How many perturbations break this model? evaluating robustness beyond adversarial accuracy." Olivier, Raphael, and Bhiksha Raj. International Conference on Machine Learning. PMLR, 2023.
   [2] "Probabilistically robust learning: Balancing average and worst-case performance." Robey, Alexander, et al. International Conference on Machine Learning. PMLR, 2022.
   [3] "Exploring the Adversarial Frontier: Quantifying Robustness via Adversarial Hypervolume." Guo, Ping, et al. arXiv preprint arXiv:2403.05100 (2024).

2. While the paper presents test results on the original test samples on CIFAR-10 in Table 9,  it does not provide results for the ImageNet dataset. The authors should explain this choice and consider including ImageNet results for a more comprehensive comparison.

**Limitations:**

See weakness and questions.

---

> ### Author Rebuttal · Authors · 2024-08-07
>
> We express sincere gratitude for your constructive and detailed comments.
>
>
> ## Weakness 1:  Definition of adversarial robustness； Algorithm in appendix reduces coherence; Clarification on Figure 2.
>
>
>
> （1） We apologize for the lack of  definition. To clarify, adversarial robustness evaluation refers to the process of assessing a model's resilience against adversarial attacks, which are inputs intentionally designed to deceive the model. We will include it  in the next version .
>
> （2） Due to space limitations, we placed the algorithm in the appendix. However, we understand your concern about coherence and will move it to the main text in the next version.
>
> （3）To clarify how RA and the GREAT Score are integrated into Figure 2, we provide the additional details:
>
> 1. Auto-Attack RA: RA at each perturbation level is the fraction of correctly classified samples under perturbation. GREAT Score RA: Cumulative RA at each perturbation level is the fraction of samples with GREAT Scores above that level.
>
> 2. Blue curve: RA from empirical Auto-Attack. Orange curve: RA from GREAT Score, providing a certified robustness guarantee.
>
> 3. Trend similarity suggests GREAT Score reflects empirical robustness. The gap indicates the difference between certified and empirical robustness, not necessarily GREAT Score's inferiority, but possible undetected adversarial examples at higher perturbations.
>
>
>
>
>
> ## Weakness 2: Unknown data distribution.
>
>
>
>
> Thank you for your insightful comment. By "unknown" data distribution, we mean the true data distribution is unknown (e.g., the data distribution of all nature images). In practice, one trains a generative model to learn this unknown function. The aim of the generative model to serve as a proxy for the true data distribution. In the Appendix A.2, we provide an explanation of how the generative model can effectively match the true data distribution. This justification supports our claim that the proposed metric can scale to unknown data distributions, which is also an aim for current state-of-the-art image generative models.
>
>
>
> ## Weakness 3: No guarantee on producing benign samples.
>
>
>
> We acknowledge that the performance of our metric depends on the generative model's ability to produce benign samples. Our approach assumes the use of high-quality GANs.
>
> - **Ablation Study**: As demonstrated in our ablation study, using GANs with better generative quality improves the ranking coefficient, indicating stronger performance of the robustness metric.
> - **Theoretical Support**: In Appendix 2, we discuss how GANs can provably match the data distribution, providing a theoretical foundation for their use in our approach.
>
>
>
>
>
>
> ## Weakness 4: Training and online learning for Generative Models.
>
>
> Thank you for your valuable suggestion. Our ablation study indicates that improving the quality of the generative model significantly enhances the ranking correlation. Therefore, we definitely need better generative models. Currently, we are using off-the-shelf generative models, but we will consider incorporating online learning techniques to further improve their performance.
>
>
>
> ## Weakness 5: Limitation to white-box settings.
>
> Thank you for highlighting this important distinction. However, we would like to clarify that our discussion on limitations pertains specifically to the evaluation of attack-independent robustness, which we stated is limited to white-box settings. We acknowledge that adversarial accuracy can indeed be assessed in black-box scenarios, as demonstrated by methods like Square Attack.
>
>
>
>
> ## Question 1: Omission of works on robustness evaluation.
>
>
> Thank you so much for recommending the related works we need , we will add them in next version. Below is the added content.
>
>
> In addition to discussed works, several studies evaluate model robustness differently. [Olivier 2023] introduce adversarial sparsity, quantifying the difficulty of finding  perturbations, providing insights beyond adversarial accuracy.  [Robey et al. 2022] propose probabilistic robustness, balancing average and worst-case performance by enforcing robustness to most perturbations, better addressing trade-offs. [Guo et al. 2024] introduce the adversarial hypervolume metric, a comprehensive measure of robustness across varying perturbation intensities.
>
>
> [1] "How many perturbations break this model? evaluating robustness beyond adversarial accuracy." Olivier, Raphael, and Bhiksha Raj. International Conference on Machine Learning. PMLR, 2023.
> [2] "Probabilistically robust learning: Balancing average and worst-case performance." Robey, Alexander, et al. International Conference on Machine Learning. PMLR, 2022.
> [3] "Exploring the Adversarial Frontier: Quantifying Robustness via Adversarial Hypervolume." Guo, Ping, et al. arXiv preprint arXiv:2403.05100 (2024).
>
>
> ## Question 2: Results on the original test samples on ImageNet.
>
>
>
> Following the reviewer's suggestion, we conducted an additional experiment using 500 test samples from the original ImageNet dataset, with image dimensions of 224x224 pixels, and reported their GREAT Scores in the table below. However, it is important to note that unless these test samples are generated by a generative model, they may not fully represent the data distribution captured by such a model. The correlation between the GREAT Score ranking and the RobustBench ranking remains consistent with the results observed on the generated samples.
>
>
> | Model  | RobustBench Accuracy (%) | AutoAttack Accuracy (%) | GREAT Score |
> |------------|---------------------------|-------------------------|-------------|
> | Trans1 | 38.14                     | 56.0                   | 0.58    |
> | Trans2 | 34.96                     | 50.2                 | 0.48      |
> | LIBRARY | 29.22                    | 50.6                  | 0.49      |
> | Fast   | 26.24                     | 39.4                    | 0.41     |
> | Trans3 | 25.32                     | 39.8                   | 0.32      |

---

> > ### Comment · Reviewer_P6PA · 2024-08-09
> >
> > Dear Authors,
> >
> > Thank you for submitting your rebuttal in response to the concerns I previously raised. I appreciate the effort you have put into addressing these issues. Please find below my comments following your rebuttal:
> >
> > ## Overview
> > The manuscript presents an intriguing concept in the evaluation of adversarial robustness, which offers a novel perspective that diverges from the traditional attack-based methods. This is a commendable approach that contributes to the field.
> >
> > Nevertheless, I observe that the proposed inference-based method bears resemblance to existing work in robustness evaluation, such as Adversarial Sparsity. While the incorporation of generative models mitigates some of the limitations inherent in prior methods, it also introduces new challenges associated with the characteristics of GANs. These issues are likely to be of interest to the readers.
> >
> > Overall, these limitations do not hurt the significance of your contribution. Furthermore, your response has addressed most of my concerns, so I decide to raise my score. However, I have summarized your response and identified several additional points that may warrant further consideration.
> >
> >  - **Presentation:** I apprecite your efforts in improving the readability, no further questions.
> >  - **Unknown Data Distribution:** The discussion surrounding the concept of a 'true unknown data distribution' is somewhat controversial. The use of training data for the model under evaluation in the training of generative models raises questions about the definition of unknown data distribution. This issue is not present in studies focusing solely on generative models.
> >    - What would be the implications if the training data were not available for GAN training?
> >    - How would the use of only test data affect the training, and could this potentially lead to the collapse of GANs due to a shift in distribution?
> >  - **GAN Producing Benign Examples:** This may be alleviated by considering the convergence properties of GANs. However, same issue of data dependence persists.
> >  - **Online Learning:** The topic of online learning is a potential avenue for future research by the authors and shares the aforementioned limitation regarding unknown data distribution. This is particularly relevant given the requirement for substantial data volumes.
> >  - **Minor Issues:**
> >    - **Black-box Setting:** The extension to a black-box-based accuracy appears to be a minor one, yet it should be acknowledged in the manuscript.
> >    - **Related Works:** I note that the authors intend to include additional related works in the revised manuscript. I believe that a more in-depth discussion of these works could provide valuable insights for future research in this area.
> >    - **Additional Results:** I am pleased with the results provided and have no further concerns in this regard.
> >
> > I look forward to seeing the final version of the manuscript and wish to see your continued research in this important area.
> >
> > Best regards,

---

> > > ### Author Response · Authors · 2024-08-09
> > >
> > > Dear Reviewer,
> > >
> > >
> > > Thank you very much for your insightful and constructive feedback on our manuscript. We are pleased to learn that the reviewer has raised the score. We have carefully considered your comments and have made the following revisions to address your concerns:
> > >
> > > - Presentation and Related Works: Thank you for your suggestion, especially for pointing out the 3 important related papers, we will update this in the final version.
> > > - Unknown Data Distribution:
> > > On the review comment "What would be the implications if the training data were not available for GAN training?". We are unsure what the review meant, could the reviewer elaborate on this point? We note that our framework uses an off-the-shelf generative model for robustness evaluation, which does not involve the training of a generative model.
> > > On the review comment "How would the use of only test data affect the training, and could this potentially lead to the collapse of GANs due to a shift in distribution?" We are also unsure about this question. Test data should not affect the training by any means.
> > > - GAN producing benign examples: We agree that the GAN Producing Benign Examples rely heavily on the generative quality to approximate the true distribution, as already justified by our results in Figure 3. We also want to emphasize that our framework is not limited to GAN. We also include diffusion models in our results.
> > > - Online Learning: We agree with the reviewer that this is a good future direction to explore.
> > > - Minor issues:
> > > Black box setting: We thank the reviewer for pointing this out, we will acknowledge it in the manuscript.
> > > Related Works: We will ensure we include sufficient discussion of related works.
> > > Additional Results: We thank you for pointing out that we should also add the Imagenet experiment, we will include it in the appendix of the final version.
> > >
> > > Thanks again for your efforts.
> > >
> > > Sincerely,

---

> > > > ### Comment · Reviewer_P6PA · 2024-08-10
> > > >
> > > > Dear Authors,
> > > >
> > > > Thank you for your esponse to my previous comments regarding the training data.
> > > >
> > > > I would like to elaborate on my perspective concerning the training data for GANs. I have observed that your framework employs an off-the-shelf model, which has been pre-trained on a dataset comprising natural images. This approach aligns with the conventional focus of most research on generative models, which predominantly centers on the **natural image manifold**.
> > > >
> > > > Nonetheless, I would like to draw attention to scenarios such as when the input and output images are not of natural scenes, for instance, medical X-ray images. In such cases, it may be necessary to train a specialized model. This consideration leads to the question of how one might acquire an appropriate dataset to ensure that the GAN is capable of generalizing across different models' evaluations.
> > > >
> > > > While these concerns may not hold significant weight in discussions limited to natural images, I believe it is crucial to acknowledge this limitation. My intention is to highlight this aspect so that it may be addressed in future research endeavors, thereby examining the impact of data distribution on the performance of generative models.
> > > >
> > > > Warm regards,

---

> > > > > ### Author Response · Authors · 2024-08-10
> > > > >
> > > > > We thank the reviewer again for the insightful discussion. We agree that a specialized generative model is needed when we are evaluating domain-specific image classifiers. One good example is Section 4.5, where we use InterFaceGAN (a domain-specific image generator) to evaluate the robustness of online facial recognition APIs. We will make sure this point and the reviewer's comment are reflected in the revised version of our paper.

---

### Official Review · Reviewer_D3uH · 2024-07-11

**Soundness:** 3
**Presentation:** 3
**Contribution:** 3
**Rating:** 6
**Confidence:** 3

**Summary:**

The authors propose the GREAT score that uses conditional generative models to mimic the data generating distribution. Thereafter, the classification margin on a set of generated images can be used to obtain a global robustness score. For this, the authors make the connection between local and global robustness explicit and show that the classification margin yields a lower bound on the robustness w.r.t. L2 perturbations (convolution with Gaussian). The authors empirically validate their GREAT score on various benchmarks and highlight the strong correlation to other common robustness evaluations while reducing the computational cost for the robustness evaluation significantly.

**Strengths:**

1. The authors propose an efficient procedure to rank the robustness of models using generative models
1. GREAT can be used with "off-the-shelf" generative models and does not require specialized training etc.
1. GREAT does not require access to the gradient
1. The paper is well-written and easy to follow

**Weaknesses:**

1. There are several assumptions on the generative model that are not sufficiently/prominently enough covered. The assumptions are: (a) the model generates an instance actually belonging to the conditioned class, (b) the true class is unambiguous (e.g., conversely, there might be cases where the "Bayes optimal" model cannot decide between two or more classes). (c) the generative model is a good approximation of the true data-generating distribution. The authors should highlight such limitations more and their implications for the guarantees/method.
1. Since the authors emphasize the guarantee on the average robustness of a model, the authors could elaborate more on the practical importance of such a guarantee
1. The derived Lipschitz constant might be a loose estimate since the Lipschitz constant also includes the generative model and not only the neural network. This is not accurately reflected in, e.g., Eq 10. Here it seems the model was convolved with the Gaussian ($g' * N(0,1))$), but it should actually be $((g' \circ G) * N(0,1))$.

Minor:
- The font size in figures and tables is very small

**Questions:**

1. Figure 2 shows a large gap between empirical attacks (upper bound) and the GREAT score (lower bound). To what extent do the authors expect this gap to be due to the looseness of the respective upper and lower bounds?
1. How is the class label (input of conditional generative model) distributed in the experiments for calculating the GREAT score?
1. Would it also be possible to derive guarantees for a subset of the data distribution's support? For example, obtaining class-specific average robustness guarantees?

**Limitations:**

The authors sufficiently addressed the limitations (except for the weaknesses stated above).

---

> ### Author Rebuttal · Authors · 2024-08-07
>
> We appreciate your detailed and constructive comments, and we are encouraged that you find our work “well written, easy to follow”.
>
>
> ## Weakness 1: Generative model limitations: (a) valid class instance, (b) unambiguous class, (c) data approximation.
>
>
> Thank you for your feedback regarding the assumptions on the generative model. We would like to specify how to ensure these points:
>
> (a) We acknowledge that the performance of our metric relies on the generative model's capability to produce samples belong to the conditioned class. Recent studies ([Nicolas, 2021] and [Tengyuan Liang, 2021]) have shown GANs' convergence to true data distributions under specific conditions. Moreover, Figure 3 shows high-quality instances produced by the generative models, as evidenced by the inception score and the Spearman's rank correlation between the GREAT Score and RobustBench.
>
>
> (b) We recognize that there may be instances where the true class is ambiguous, potentially impacting the performance. Given that our focus is on evaluating the robustness of classifiers, it is important to note that the labels we use are typically well-defined and distinctive. When two classes exhibit ambiguity, determining their robustness is a separate issue that needs to be addressed before applying our  method. Consequently, we consider the issue of label ambiguity to be outside the scope of our method.
>
> (c) We understand that the assumption of the generative model being a good approximation of the true data-generating distribution is crucial. Recent work has demonstrated the convergence rate of approaching the true data distribution for a family of GANs under certain conditions. Please refer to Appendix A.2 for more details.
>
>
> References:
>
> Nicolas Schreuder, Victor-Emmanuel Brunel, and Arnak Dalalyan. Statistical guarantees for generative models without domination. In Algorithmic Learning Theory, pages 1051–1071. PMLR, 2021. 14
>
> Tengyuan Liang. How well generative adversarial networks learn distributions. The Journal of Machine Learning Research, 22(1):10366–10406, 2021. 14
>
>
>
> ## Weakness 2: Importance of robustness guarantee.
>
>
>
> We appreciate your feedback regarding the importance of our score. Our method is a form of certified robustness, a concept well-utilized by [Tramer, F. 2020] and [Pintor, M. 2022], and recognized as a secure way for validating defenses. Many empirical defenses were soon to be broken by advanced attacks because they were not certified to be robust. Additionally, our score can be used for robustness ranking across different models, as discussed in our paper. This ranking allows for comparison of the resilience of models against adversarial attacks.
>
>
> 1. Tramer, F., Carlini, N., Brendel, W., & Madry, A. (2020). On adaptive attacks to adversarial example defenses. Advances in neural information processing systems, 33, 1633-1645.
>
> 2. Pintor, M., Demetrio, L., Sotgiu, A., Demontis, A., Carlini, N., Biggio, B., & Roli, F. (2022). Indicators of attack failure: Debugging and improving optimization of adversarial examples. Advances in Neural Information Processing Systems, 35, 23063-23076.
>
>
> ## Weakness 3: Equation Clarification.
>
> Thank you for your insightful comment. We would like to clarify that, as defined in Equation 9, $g'$ is inherently linked with the Gaussian distribution through the generative model $G(\cdot)$. Therefore, the formulation $(g' \circ G) * N(0, I)$ is indeed consistent with our definitions. The inclusion of $g'$ with the Gaussian reflects the model's behavior in the context of our robustness analysis. We believe this captures the relationship and does not undermine the integrity of our Lipschitz constant estimation.
>
> ## Weakness 4: Small fonts.
>
> We appreciate your feedback regarding the small font size. We will increase the font size in the next version to enhance readability.
>
>
>
>
> ## Question 1: Gap between empirical attacks and GREAT score?
>
> We thank the reviewer for bringing up this discussion. We believe your concern can be addressed by comparing empirical versus certified robustness. AutoAttack does not guarantee the absence of undiscovered adversarial examples when it fails, a common limitation of empirical robustness evaluation. In contrast, our score provides a certified robustness guarantee, ensuring no adversarial examples exist within the certified perturbation range. Therefore, the gap between our certified curve and AutoAttack's empirical curve does not imply our method's ineffectiveness; it could indicate undiscovered adversarial examples at higher perturbation radii. Without sound and complete attacks that confirm no adversarial examples exist if they fail, we cannot rule out the possibility of hidden adversarial examples in these high-radii regimes, as highlighted by certified robustness analysis.
>
>
>
> ## Question 2: Label distribution?
>
>
> We appreciate the reviewer's insightful idea. We utilize a conditional generative model to generate samples based on class labels. The class labels are uniformly distributed.
>
>
>
> ## Question 3: Class-specific robustness?
>
>
> Thank you for the insightful question. We acknowledge that deriving guarantees for subsets of the data distribution is important to explore.
>
> Our current work focuses on evaluating model robustness over the entire data distribution. Given our capability to perform class-specific generation, we can indeed measure class-specific robustness guarantees. By generating samples specific to each class, we can evaluate how robust the model is against adversarial attacks targeting particular classes, thereby evaluating  class-specific robustness .
>
> Actually, we have performed a similar analysis to evaluate group-level robustness for facial recognition. This method can be extended to derive class-specific robustness guarantees. Table 4 in our paper displays the group-level GREAT Score results. Our evaluation reveals interesting observations: Most APIs  exhibit a large discrepancy in scores between "Old" vs. "Young" groups.

---

> > ### Comment · Reviewer_D3uH · 2024-08-08
> > **Response to rebuttal**
> >
> > I thank for the detailed explanations. I will raise the score to 6 if the authors will include a detailed discussion concerning Weakness 1 in the main part of the revised paper and "correct" the RHS of Eq 10.

---

> > > ### Author Response · Authors · 2024-08-08
> > >
> > > Thank you for your valuable feedback. We are delighted to learn that the reviewer is inclined to raise the score. While we can't edit the submission at this point, we will include a detailed discussion of Weakness 1 in the main part of the revised paper, specifically in the experimental section, with a dedicated discussion on GANs. Additionally, we will explicitly add the dependency of the generator $G$ in the right-hand side of Equation 10 as suggested.

---

### Official Review · Reviewer_kqpq · 2024-07-13

**Soundness:** 3
**Presentation:** 3
**Contribution:** 3
**Rating:** 6
**Confidence:** 3

**Summary:**

The paper introduces a novel framework called GREAT Score (Global Robustness Evaluation of Adversarial Perturbation using Generative Models), aimed at evaluating the global robustness of machine learning models against adversarial perturbations. Unlike traditional methods that aggregate local robustness results from a finite set of data samples, GREAT Score leverages generative models to approximate the underlying data distribution, providing a more comprehensive global robustness assessment.

**Strengths:**

- The GREAT Score framework introduces a novel method for global robustness evaluation using generative models, which is a fresh and innovative approach in the field.
- The paper provides solid theoretical foundations with formal definitions, probabilistic guarantees, and detailed proofs, enhancing the credibility of the proposed method.
- The GREAT Score framework offers significant computational savings over traditional methods and can audit privacy-sensitive black-box models without accessing real data, highlighting its practical importance and broad applicability.

**Weaknesses:**

1. The GREAT Score in the paper primarily focuses on adversarial perturbations under the L2 norm. While this is a common setting in adversarial attack research, it lacks ablation studies for other norms, such as the L∞ norm
2. The GREAT Score framework relies on generative models (such as GANs or diffusion models) to approximate the true data distribution. If the quality of the generative model is not high, the generated samples may not accurately represent the true data distribution, thus affecting the accuracy of robustness evaluation. Besides, the evaluation results of the GREAT Score also depend on the generated sample set. If the sample set is biased or fails to comprehensively cover the diversity of the data distribution, the evaluation results may be inaccurate or unrepresentative.
3. The evaluation of online facial recognition APIs using GREAT Score is innovative, but the paper could provide more detailed analysis and discussion on the specific challenges and insights derived from this application. For instance, exploring the variability in robustness scores among different groups (e.g., age, eyeglasses) in greater depth and providing potential reasons for these variations would add depth to the analysis.
4. The calibration process described in Section 3.5 appears somewhat ad-hoc, relying on grid search for optimizing temperature parameters. This could be perceived as lacking robustness and generalizability. A more systematic approach to calibration, possibly incorporating advanced optimization techniques or sensitivity analysis, would strengthen the framework. Discussing the stability and consistency of the calibration process across different models and datasets would also be beneficial.
5. Despite claiming computational efficiency, the paper does not provide a detailed analysis of the scalability of the GREAT Score framework, especially in the context of extremely large datasets and models. A thorough examination of how the computation time scales with increasing data size and model complexity would add significant value. This could include empirical results demonstrating the method's performance on larger datasets or theoretical analysis of its computational complexity.

**Questions:**

1. Have you considered extending the GREAT Score framework to other norm-based perturbations like L1 or L∞ norms? If so, what are the potential challenges or theoretical adjustments needed?
2. How does the quality of the generative model affect the GREAT Score evaluation? Have you conducted any experiments using generative models of varying quality to analyze this impact?
3. Can you provide more detailed information on the scalability of the GREAT Score framework with respect to extremely large datasets and models? How does the computation time scale with increasing data size and model complexity?
4. How stable and consistent is the calibration process described in Section 3.5 across different models and datasets? Have you explored any advanced optimization techniques for calibration?

**Limitations:**

Yes

---

> ### Author Rebuttal · Authors · 2024-08-07
>
> We appreciate your detailed and constructive comments.
>
>
>
> ## Weakness 1 & Question 1: Ablation studies for other norms (e.g., L∞)?
>
>
> We thank for bring the limititions of our theorem. As stated in Section 5, our framework currently focuses on the $\mathcal{L}_2$ norm due to limitations in extending Stein's Lemma to other $\mathcal{L}_p$ norms (see Lemma 4 in Appendix A.3). We agree that generalizing beyond $\mathcal{L}_2$ robustness would be beneficial, but it is challenging without a generalized Stein's Lemma for other $\mathcal{L}_p$ norms. As of now, we are not aware of any results that support such an extension.
>
>
>
> ## Weakness 2 : Generative model quality and sample set bias.
>
>
> The reviewer's intuition is correct. In our ablation study of generative models (GMs), we do find that better GMs give higher ranking. In Figure 3, we show that increasing the Inception Score of GMs can significantly increase the Spearman's rank correlation. Intuitively, a higher inception score means better learning of GMs to the underlying data distribution, resulting in improved ranking efficiency in our case. Recent works such as [Nicolas, 2021] and [Tengyuan Liang, 2021] have proved theoretically that GANs can learn the true generating distribution under certain conditions.  In summary, the effectiveness of GREAT Score is positively correlated with the generation capabilities of the GM in use. Please refer to Appendix A.2 for details
>
> References:
> Nicolas Schreuder, Victor-Emmanuel Brunel, and Arnak Dalalyan. Statistical guarantees for generative models without domination. In Algorithmic Learning Theory, pages 1051–1071. PMLR, 2021. 14
>
> Tengyuan Liang. How well generative adversarial networks learn distributions. The Journal of Machine Learning Research, 22(1):10366–10406, 2021. 14
>
>
> ## Weakness 3: Detailed analysis on group variability.
>
>
>
> We appreciate your suggestion to provide a more detailed analysis among different demographic groups to add depth to our analysis.
>
> Exploring robustness variability among demographic groups is crucial for mitigating biases in facial recognition systems. Studies like [Klare, 2012] and [Deb, 2020] have shown that factors such as age and race can affect recognition accuracy.
>
>
> We have found an evidence for the facial recognition system has less elder faces in training data.  [Meade, R. 2021] found that the majority of models that perform gender classification are trained on the most popular actors and actresses sourced from Wikipedia and IMDB. As this is a celebrity dataset, most celebrities appear much younger than someone of the same age who is not a celebrity - creating a bias towards classifying older people.
>
>
>
>
>
>
> We will add this discussion in the revised version.
>
> References:
>
> [1] Klare, Brendan F., et al. "Face recognition performance: Role of demographic information." *IEEE Transactions on Information Forensics and Security*. 2012.
>
> [2] Deb, Debayan, et al. "Face Recognition Performance: Role of Demographic Information on Consumer- to Organizational-Level Applications." *IEEE Transactions on Information Forensics and Security*. 2020.
>
> [3] Meade, R., Camilleri, A., Geoghegan, R., Osorio, S., & Zou, Q. (2021). Bias in machine learning: how facial recognition models show signs of racism, sexism and ageism.
>
>
> ## Weakness 4 & Question 4: Stability of grid search in calibration ?
>
> Thank you for your feedback on the calibration process. While grid search for optimizing temperature parameters might seem ad-hoc, it is widely used and accepted in state-of-the-art research.
>
> Several recent studies have used grid search for calibration due to its simplicity and effectiveness. [Guo et al., 2017] optimize temperature scaling for neural network outputs, forming a foundational reference for modern calibration techniques. Similarly, [Kumar et al., 2019] use grid search to optimize calibration parameters, demonstrating its utility for reliable uncertainty estimates.
>
> Grid search is simple and effective under time constraints. Advanced techniques require significant resources and may not yield much better results. A systematic study with these techniques could strengthen our framework for future work.
>
> References:
>
> [1] Guo, Chuan, et al. "On calibration of modern neural networks." *International Conference on Machine Learning*. 2017.
>
> [2] Kumar, Ananya, et al. "Verified uncertainty calibration." *Advances in Neural Information Processing Systems*. 2019.
>
>
>
>
> ## Weakness 5 & Question 3: Scalability and computational efficiency on large datasets/models?
>
> To address the question on the scalability of the GREAT Score framework, we conducted experiments using three ResNets, varying dataset sizes from 500 to 2000 images. We recorded the computation times in ms without employing any attack mechanisms.
>
>
> | Dataset Size  | ResNet50  | ResNet101  | ResNet152  |
> |-------------------|---------------|----------------|----------------|
> | 500               | 3274       | 6251       | 9149        |
> | 1000              | 6529       | 12528      | 18339       |
> | 1500              | 9785       | 18838      | 27481       |
> | 2000              | 12960      | 24917      | 36588       |
>
> The results show a linear increase in computation time with increasing dataset size and model complexity. More complex models like ResNet152 took proportionally more time than simpler ones like ResNet50.
>
> This linear scalability demonstrates that the GREAT Score  efficiently handles larger datasets and more complex models, making it suitable for large-scale applications.
>
>
>
> ## Question 2: Impact of generative model quality ?
>
>
> Yes, we thank the reviewer for pointing out this question, actually, we have done an Ablation study on GANs and DMs. Evaluating on CIFAR-10, Figure 3 compares the inception score (IS) and the Spearman’s rank correlation coefficient between GREAT Score and RobustBench on five GANs and DDPM. One can observe that models with higher IS attain better ranking consistency.

---

> > ### Comment · Reviewer_kqpq · 2024-08-13
> >
> > Thanks for your clarification. If more search methods can be used as an ablation study, I will raise my score from 5 to 6.

---

> > > ### Author Response · Authors · 2024-08-13
> > >
> > > We greatly appreciate your insightful feedback and the suggestion for this valuable ablation study. We agree that demonstrating our framework's reliability across different search methods is crucial. We are also very pleased to learn that you are inclined to raise the score based on these additions.
> > >
> > > In response to your recommendation, we have conducted a comprehensive ablation study comparing various search methods for calibration on the temperature, including grid search, binary search, simulated annealing, and genetic algorithm. This study was performed under the same experimental settings as the calibration process described in our paper. We will incorporate these findings into Section 4.4 of the final version, alongside the ablation study of generative models.
> > >
> > >
> > >
> > >
> > >
> > >
> > > The results of our experiment are summarized in the table below:
> > >
> > >
> > >
> > > | Method | GREAT Score vs. RobustBench Correlation | GREAT Score vs. AutoAttack Correlation | Best Temperature |
> > > | ------ | -------------------------------------- | -------------------------------------- | ---------------- |
> > > | Grid Search | 0.8971 | 0.6941 |0.00742 |
> > > | Binary Search | 0.8971 | 0.6941 | 0.00781 |
> > > | Simulated Annealing | 0.8971 | 0.6941 | 0.00748 |
> > > | Genetic Algorithm | 0.8971 | 0.6941 | 0.00879 |
> > >
> > >
> > > For the implementation of simulated annealing and genetic algorithm, we ensured fair comparison by maintaining consistent search space ([0, 2]), precision (minimum step size of 0.00001), and computational resources across all algorithms.
> > >
> > > Simulated Annealing:
> > > We implemented a standard simulated annealing algorithm with an exponential cooling schedule. The algorithm starts with an initial temperature of 1 (this temperature is not the same as the temperature in calibration) and employs an exponential cooling schedule with a decay rate of 0.95, continuing until the temperature reaches a minimum of 0.001 or other termination criteria are met. To accommodate time constraints, we reduced the number of iterations in the inner loop to 10, while maintaining the overall algorithm structure. The initial temperature was set high to allow for broad exploration, gradually decreasing to enable fine-tuning.
> > >
> > > Genetic Algorithm:
> > > Our genetic algorithm implementation used a population of 30 candidate solutions over 50 generations. We employed tournament selection, single-point crossover, and a mutation rate of 0.3. The population size and number of generations were adjusted to align with the computational time of the simulated annealing algorithm.
> > >
> > > Both algorithms were adapted to optimize the temperature parameter in our GREAT Score calibration process, with the objective of maximizing the correlation scores. While the reduced iterations might affect the algorithms' ability to find the global optimum, this trade-off reflects real-world constraints and allows for a fair comparison of their performance under limited computational resources.
> > >
> > >
> > > The ablation study demonstrates that while different search methods (grid search, binary search, simulated annealing, and genetic algorithm) yield slightly different optimal temperature values, they all converge to the same ranking results. This consistency across different search methods highlights the reliability and practical applicability of the GREAT Score calibration process.

---

> > > > ### Comment · Reviewer_kqpq · 2024-08-14
> > > >
> > > > I'm grateful for your additional experiments. Based on the results, different search algorithms don't have a significant impact, which addresses my concern. From this perspective, whether to include this ablation study in the final paper is up to you. Anyway, I will improve my score and advocate for this paper to be accepted.

---

> > > > > ### Author Response · Authors · 2024-08-14
> > > > >
> > > > > We thank the reviewer for the prompt response and for your support in championing our paper!

---

### Official Review · Reviewer_gkrf · 2024-07-15

**Soundness:** 2
**Presentation:** 3
**Contribution:** 2
**Rating:** 3
**Confidence:** 4

**Summary:**

The paper addresses the important and under-explored problem of "global robustness evaluation" for neural networks. It proposes GREAT Score, a novel framework for assessing global robustness using generative models (GMs).  Besides, through Monte Carlo sampling from GMs and using Hoeffding's concentration bound, the algorithm can reach an epsilon probabilistic guarantee on the sample mean's closeness to the true mean. The paper then applies their proposed algorithm on various classifiers using GMs to measure global robustness scores.

**Strengths:**

1) The paper attempts to tackle a significant gap in global robustness assessment, offering a reasonable and innovative contribution to the field.
2) The paper is well-organized, clearly written, and easy to follow.
3) The experimental results show high consistency between GREAT Score and attack-based model rankings on RobustBench, demonstrating its potential as an efficient alternative to existing robustness benchmarks.

**Weaknesses:**

1) The reliance on GANs as a proxy for the true data distribution raises concerns about the method's accuracy. To the best of my knowledge, current GANs do not generate better coverage than the test set. GANs are a bad estimation of the underlying data distribution with known issues such as bias and model collapsing. Considering model collapse, the fixed test set is likely to have even better distribution coverage than the samples generated from GAN. It would be much more reliable and convincible by involving  the recent class generative models.
2) I also encourage the authors to include experiments with other local robustness estimators, further strengthening the submission.
How does the choice of generative model and local robustness estimator affect the reliability of the global measure computed by the paper?
3) Theoretically, while the authors provide a probabilistic guarantee on the obtained estimates derived from GMs and true estimate, there's a lack of theoretical bound on gap between the true estimate and models' global robustness arising from the distance of the  generative distribution and underlying data distribution. Otherwise, the significance and utility of the GREAT score is unclear for me, and this omission makes it unclear how the accuracy of the empirical distribution affects the overall error, beyond just sample complexity.

**Questions:**

Please respond my questions in the weakness part.

**Limitations:**

Yes

---

> ### Author Rebuttal · Authors · 2024-08-07
>
> We express sincere gratitude for your valuable feedback and constructive comments.
>
> ## Question 1: GAN reliability and distribution coverage?
>
> Thank you for your feedback on the reliance on GANs as a proxy for the true data distribution. We acknowledge the concerns about the method's accuracy, particularly the issues of bias and model collapse inherent in GANs. We would like to address these concerns and explain our approach in detail.
>
> Firstly, we would like to clarify that we utilized conditional generative models in our framework. Conditional generative models, such as Conditional GANs (cGANs), allow for more controlled generation of samples by conditioning on specific labels or attributes. This helps in generating more representative and varied samples, which can mitigate some of the biases and issues associated with traditional GANs.
>
> Recent works such as [Nicolas, 2021] and [Tengyuan Liang, 2021] have proved theoretically that GANs can learn the true generating distribution under certain conditions. Besides GANs, we have diffusion models in our analysis, and as the reviewer expected, it gives a better robustness analysis, as shown in Fig. 3. Please also see Appendix A.2 for details.
>
>
>
> References:
>
> Nicolas Schreuder, Victor-Emmanuel Brunel, and Arnak Dalalyan. Statistical guarantees for generative models without domination. In Algorithmic Learning Theory, pages 1051–1071. PMLR, 2021. 14
>
> Tengyuan Liang. How well generative adversarial networks learn distributions. The Journal of Machine Learning Research, 22(1):10366–10406, 2021. 14
>
>
> ## Question 2: Experiments with other estimators?
>
>
> Thank you for the insightful question. Following the reviewer's suggestion, in our experiments below, we included two types of local robustness estimators to evaluate model robustness more comprehensively: **Input Gradient Norm (as a proxy of sensitivity)** and **Entropy of the classifier output (as a proxy of confidence)**. We note that unlike our proposed GREAT Score, these estimators do not provide any certified robustness guarantee. Here's why we chose these estimators and how they affect our global measure:
>
>
>
> 1. **Input Gradient Norm**:
>     - **Rationale**: This estimator measures the sensitivity of the model to small perturbations in the input by calculating the norm of the gradient of the loss with respect to the input. A higher gradient norm indicates greater sensitivity to input changes, suggesting lower robustness.
>     - **Reference**: The use of input gradient norm for robustness evaluation is well-documented.  [ Finlay and Oberman, 2021] proposed a method using input gradient regularization to improve adversarial robustness.
>     - **Implementation**: We computed the gradient norm for input samples and took the average to represent the model's sensitivity.
>
> 2. **Entropy**:
>     - **Rationale**: Entropy of the classifier's output probabilities reflects the uncertainty of the model's predictions. Higher entropy indicates that the model is less confident in its predictions, which can be a sign of lower robustness.
>     - **Reference**: The relevance of entropy as a measure of model uncertainty and robustness is supported by [Smith and Gal ,2018], who explored various uncertainty measures, including entropy, for detecting adversarial examples .
>     - **Implementation**: We calculated the entropy of the output probabilities for input samples and took the average to represent the model's uncertainty.
>
>
>
> To validate the reliability of our global measure, we computed the Spearman correlation coefficient between our averaged estimation scores and the rankings from RobustBench. Besides the local robustness estimator, the experiment setup is the same as Table 1.
>
>
> #### Experimental Results
>
> |                                   | Input Gradient Norm | Entropy | GREAT Score Calibrated |
> |-----------------------------------|----------------------------|----------------|------------------------|
> | Correlation  with AutoAttack | -0.0956                     | 0.4500         | 0.6941                 |
> | Correlation  with RobustBench | 0.0343                      | 0.4951         | 0.8971                 |
>
> This table summarizes the correlation results for the local robustness estimators and the GREAT Score with respect to AutoAttack and RobustBench. These results demonstrate the effectiveness of  GREAT Score in evaluating model robustness and their alignment with established benchmarks.
>
>
>
>
> ### References
>
> Finlay, C., & Oberman, A. M. (2021). Scaleable input gradient regularization for adversarial robustness. Machine Learning with Applications, 3, 100017.
> Smith, L., & Gal, Y. (2018). Understanding measures of uncertainty for adversarial example detection. arXiv preprint arXiv:1803.08533.
>
>
>
>
>
>
> ## Question 3: Theoretical bound on distribution gap and its effect on  utility?
>
>
> We separated out the convergence of our edtimate to the true estimate versus the learned data distribution to the true data distribution because the latter has been proved in the literature. Recent works such as [Nicolas, 2021] and [Tengyuan Liang, 2021] have proved theoretically that GANs can learn the true generating distribution under certain conditions. Besides GANs, we have diffusion models in our analysis, and as the reviewer expected, it gives a better robustness analysis, as shown in Fig. 3.
>
> The reviewer's intuition is correct. In our ablation study of generative models (GMs), we do find that better GMs give higher ranking. In Figure 3, we show that increasing the Inception Score of GMs can significantly increase the Spearman's rank correlation. Intuitively, a higher inception score means better learning of GMs to the underlying data distribution, resulting in improved ranking efficiency in our case. We also had a discussion on the approximation guarantee of some GMs to the true data distribution in Sec. 6.2. In summary, the effectiveness of GREAT Score is positively correlated with the generation capabilities of the GM in use.

---

### Author Rebuttal · Authors · 2024-08-07

We appreciate the valuable feedback from the reviewers . Below is a high-level summary of our rebuttal, addressing the major concerns raised:

###  Performance and reliability of the generative model




* **Concern:** The reviewers were concerned about the dependency of our metric on the generative model's ability to produce samples that truly belong to the conditioned class, and whether generative models can appoximation to true data distribution. Besides , the reviewers also wonder theoretical bound on distribution gap and its effect on utility
* **Response:** We have provided related liteature proving the convergence rate of GANs in approximating true data distributions. We have also mentioned more details can be found in Appendix A.2.  We have reiterated the positive correlation between generative model quality and GREAT Score effectiveness, supported by our ablation study shown in Figure 3.


### Experimental Validation

* **Concern:** The reviewers requested additional evidence to support the effectiveness of our approach on Imagenet, scalability to large datasets/models, and comparison with other estimators.
* **Response:** We have provided further experimental results and comparisons with local estimators to demonstrate the robustness and reliability of our method. We also conducted an additional experiment on 500 test samples from the original ImageNet dataset and reported their GREAT scores, confirming the consistency of the correlation between the GREAT score ranking and the RobustBench ranking. We also perform experiments on different ResNets and dataset sizes to demonstrate the time efficiency of our method.

###  Practical Applications of Our Score.

* **Concern:** Reviewers highlighted  practical applications of our method and asked comparasion between our method and Auto-Attack.
* **Response:** We  discussed the practical significance of our score as a lower-bound guarantee of robustness to attacks, useful for robustness ranking and informed model selection.

### Methodological Improvements and Future Directions

* **Concern:** Reviewers suggested exploring other norms and discussed the scalability of our approach.
* **Response:** We acknowledged the limitation of our current framework to the L2 norm and highlighted this as a potential future research direction.

### Demographic and Class-Specific Analyses

* **Concern:** The reviewers emphasized the importance of demographic group analysis and class-specific robustness.
* **Response:** We clarified the hidden reason for different robustness among groups in gender classification. We metioned we can be extend our Score to derive class-specific robustness guarantees. We also specified the uniform distribution of class labels used in our experiments to ensure fairness and reliability.

###  Omitted Works

* **Concern:** Reviewers  recommended additional relevant works.
* **Response:**  We thanked the reviewer for recommending relevant works and committed to including them in the revised manuscript.


### Clarity and Presentation

* **Concern:** The clarity and presentation of certain sections were highlighted as areas for improvement.
* **Response:** We have revised the manuscript to enhance clarity and readability. Specific sections have been rewritten for better comprehension, and additional explanations have been included where necessary. We also clarified the definition and inclusion of the Lipschitz constant in our analysis, ensuring consistency with our framework. Feedback on font size has been acknowledged, and we committed to increasing it in the revised manuscript.



We believe that these changes have significantly strengthened our manuscript and addressed the reviewers' concerns. We hope that the paper is now suitable for publication.

---

### Decision · Program_Chairs · 2024-09-25

**Decision:**

Accept (poster)

**Comment:**

This paper is on evaluating adversarial robustness. There are three limitations of current evaluation methods:
- The evaluation is often limited to a fixed test dataset, which does not capture robustness on the global data distribution.
- The evaluation is often limited to white-box settings where input gradients are required.
- The evaluation may require lots of computation.

This paper proposes a “GREAT score” to address these three limitations. First, they use a generative model to approximate the unknown data distribution. Second, their evaluation method only requires forward passes (as shown in Eqn. (3)) and is useful when only black-box access is available. Third, the evaluation can use pretrained generative models, which reduces the computational burden. Furthermore, each data point for evaluation only requires one forward pass.

On the theory side, this paper shows that GREAT score corresponds to a lower bound on the robustness radius to L2 attacks. They also showed a formal probabilistic guarantee on the finite-sample performance of GREAT score. On the empirical side, the paper provides extensive evaluations using RobustBench and shows reductions in computation time, sometimes up to 2000 times.

The paper provides a novel approach to evaluate adversarial robustness effectively. It is fast, has theoretical guarantees, and is aligned with existing robustness evaluations based on the experiments on RobustBench. Several reviewers commented on the clear writing of the paper.

The authors were able to address most reviewers’ concerns. Unfortunately, one of the reviewers did not engage in the discussions. However I can see that most of their concerns are appropriately addressed. The reviewers gave several valuable suggestions to improve the current evaluations, such as ablating more hyperparameter search methods in the calibration process and discussing the quality of generative models and their effect on the evaluation of robustness.